# Quantifying Body Motion Synchrony in Autism Spectrum Disorder Using a Phase Difference Detection Algorithm: Toward a Novel Behavioral Biomarker

**DOI:** 10.3390/diagnostics15101268

**Published:** 2025-05-16

**Authors:** Jinhwan Kwon, Hiromi Kotani

**Affiliations:** 1Department of Education, Kyoto University of Education, Kyoto 612-8522, Japan; hikotan@kyokyo-u.ac.jp; 2Kyoto City Child Welfare Center Clinic, Kyoto 612-8434, Japan

**Keywords:** autism spectrum disorder, body motion synchrony, phase difference detection algorithm, behavioral biomarker, diagnostic marker

## Abstract

**Background/Objectives**: Nonverbal synchrony—the temporal coordination of physical behaviors such as head movement and gesture—is a critical component of effective social communication. Individuals with autism spectrum disorder (ASD) are often described as having impairments in such synchrony, but objective and scalable tools to measure these disruptions remain limited. This study aims to assess body motion synchrony in ASD using phase-based features as potential markers of social timing impairments. **Methods**: We applied a phase difference detection algorithm to high-resolution triaxial accelerometer data obtained during structured, unidirectional verbal communication. A total of 72 participants (36 typically developing TD–TD and 36 TD–ASD) were divided into dyads. ASD participants always assumed the listener role, enabling the isolation of receptive synchrony. Four distribution-based features—synchrony activity, directionality, variability, and coherence—were extracted from the phase difference data to assess synchrony dynamics. **Results**: Compared to the TD group, the ASD group exhibited significantly lower synchrony activity (ASD: 5.96 vs. TD: 9.63 times/min, *p* = 0.0008, Cohen’s *d* = 1.23), greater temporal variability (ASD: 384.4 ms vs. TD: 311.1 ms, *p* = 0.0036, *d* = 1.04), and reduced coherence (ASD: 0.13 vs. TD: 0.81, *p* = 0.036, *d* = 0.73). Although the mean phase difference did not differ significantly between groups, the ASD group displayed weaker and more irregular synchrony patterns, indicating impaired temporal stability. **Conclusions**: Our findings highlight robust impairments in nonverbal head motion synchrony in ASD, not only in frequency but also in terms of temporal stability and convergence. The use of phase-based synchrony features provides a continuous, high-resolution, language-independent metric for social timing. These metrics offer substantial potential as behavioral biomarkers for diagnostic support and intervention monitoring in ASD.

## 1. Introduction

Autism spectrum disorder (ASD) is a neurodevelopmental condition characterized by persistent deficits in social communication and interaction, as well as restricted, repetitive patterns of behavior, interests, or activities [1]. According to the DSM-5 [1], individuals with ASD often struggle with nonverbal communication, including eye contact, gestures, facial expressions, and body language. Whereas typically developing (TD) individuals are capable of integrating both verbal and nonverbal cues to infer others’ intentions, individuals with ASD tend to rely more heavily on verbal information due to impairments in central coherence [2,3]. This often results in inefficient referential processing, such as difficulty identifying the target of a gesture or overreliance on mouth movements rather than the eyes for emotion recognition [4].

Interpersonal synchrony—particularly in body movements—reflects the dynamic structure of communication between individuals [5,6,7]. This phenomenon, known as body motion synchrony, refers to the temporal alignment of physical behaviors during communication and is increasingly recognized as a meaningful metric for evaluating interaction quality [8,9]. Previous studies have shown that synchronized movements enhance cooperation [10], create favorable impressions [11], and are positively associated with feelings of closeness [12], intimacy [13], bonding [14], and empathy [15,16]. For instance, physical synchrony is known to occur more frequently with familiar partners [17] and has been linked to enhanced positive affect in dyadic interactions [18,19]. Moreover, counselors who exhibit higher levels of body motion synchrony with clients tend to receive more favorable evaluations [20]. Thus, body motion synchrony is now viewed as a powerful and quantifiable indicator of communication quality and social attunement.

Previous studies have increasingly shown that individuals with ASD exhibit reduced frequencies of body motion synchrony compared to TD peers. Marsh et al. [21], using rocking chairs to measure synchrony between children with ASD and their caregivers, reported significantly fewer synchrony events in ASD dyads. Comparable results were found in Fitzpatrick et al.’s [22] pendulum swinging experiment, and in a communicative finger tapping task [23], where ASD participants displayed reduced synchrony and empathy relative to TD individuals. Furthermore, Isaksson et al. [24] found that ASD participants not only tapped with less synchrony but also exhibited faster and more variable timing than their TD counterparts. These findings suggest that synchrony in ASD is both less frequent and more temporally irregular; however, relatively few studies have examined such disruptions under naturalistic verbal communication settings.

In recent years, Motion Energy Analysis (MEA) has emerged as a widely used tool for quantifying interpersonal synchrony in naturalistic conversations, including studies on autism spectrum disorder (ASD) that examine real-time social dynamics during dyadic interactions. For example, studies by Georgescu et al. (2020) and Koehler et al. (2022) have successfully applied MEA to analyze movement synchrony between autistic and neurotypical individuals during semi-structured interviews and diagnostic assessments [25,26]. MEA offers several advantages: it is non-invasive, requires no physical markers or wearable sensors, and can be applied retrospectively to video recordings [27]. These features have made MEA especially appealing for clinical and ecologically valid research settings [9]. However, MEA also has several methodological limitations. As a 2D, pixel-based technique with low temporal resolution (10 fps), MEA is sensitive to video quality, camera angle, lighting conditions, and background noise [27,28,29]. More critically, it tends to capture global motion energy rather than fine-grained movement dynamics, making it difficult to distinguish between temporally aligned versus misaligned micro-movements. Moreover, MEA provides limited insight into the temporal structure of coordination—such as lag patterns, convergence strength, or variability—which are increasingly recognized as key features of nonverbal synchrony.

To address these shortcomings, the present study introduces a phase difference detection algorithm grounded in triaxial accelerometer data as a robust and temporally precise alternative to video-based methods [30,31]. Unlike MEA, this algorithm captures three-dimensional head motion with high temporal resolution and minimal environmental constraints. Building on prior work by Kwon et al. [30], the algorithm computes the instantaneous phase of acceleration signals along each axis and derives phase differences between interlocutors, thereby quantifying the temporal alignment of head movements. Critically, this method extracts distribution-based synchrony features—including the relative frequency of synchrony, mean phase difference, standard deviation (SD), and kurtosis—which together provide a multifaceted profile of interpersonal coordination. These metrics go beyond simple co-occurrence by capturing not only how often synchrony occurs, but also its temporal alignment, consistency, and convergence strength, essential for evaluating the quality of alignment. Recent studies have demonstrated the algorithm’s utility in distinguishing bidirectional vs. unidirectional verbal communication dynamics [31]. Thus, the phase difference framework enables a more nuanced, scalable, and quantitative assessment of nonverbal synchrony, making it highly suitable for both basic research and diagnostic innovation in autism studies.

The current study builds on this methodological advancement by analyzing head motion synchrony during structured, face-to-face communication between typically developing (TD) individuals and individuals with ASD. Critically, we employed a unidirectional verbal communication paradigm in which the ASD participants always assumed the listener role, thereby isolating the receptive aspect of synchrony without the confound of role-switching or expressive language production. Prior work suggests that listener behaviors—particularly backchannel feedback such as nodding or brief vocal affirmations—are tightly coupled with speaker rhythms and contribute to mutual alignment [32,33,34]. Thus, this design allowed for a precise investigation of implicit synchrony in ASD under ecologically valid yet experimentally controlled conditions. Our approach utilized triaxial accelerometers and a phase difference detection algorithm to extract high-resolution head motion data, from which we computed four statistical descriptors of synchrony: relative frequency, mean phase difference, SD, and kurtosis. These features captured both the activity level and the temporal precision of interpersonal alignment.

By comparing TD–TD dyads (TD condition) with TD–ASD dyads (ASD condition), this study aimed to achieve the following:Quantify group differences in body motion synchrony under unidirectional verbal communication;Characterize the variability and timing of listener synchrony in ASD;Assess the utility of phase-based synchrony features as potential behavioral markers for ASD.

Ultimately, this research contributes to the growing body of work linking sensorimotor timing to social cognition and offers a framework for objective, scalable, and clinically meaningful assessments of nonverbal social function in autism.

## 2. Materials and Methods

### 2.1. Participants

A total of 72 participants took part in this study, including 54 typically developing (TD) individuals and 18 individuals with autism spectrum disorder (ASD). The 54 TD participants were assigned to two different conditions: 36 participated in the TD condition, forming 18 TD–TD dyads, and the remaining 18 were assigned to the ASD condition, where each TD participant was paired with an ASD participant to form 18 TD–ASD dyads. In total, 36 dyads were created for the experiment. These two groups of TD participants were mutually exclusive, and no individual participated in both conditions. All TD participants were undergraduate students at Kyoto University of Education. The mean age of the TD group was 21 years (SD = 0.92), and that of the ASD group was 23 years (SD = 1.65). All participants were native Japanese speakers with normal or corrected-to-normal vision and normal hearing. To control for the potential effects of interpersonal familiarity on body motion synchrony [12,14,17], each dyad was composed of two same-gender individuals who had no prior acquaintance. Gender ratios were strictly balanced (1:1) across all conditions.

Inclusion criteria for the ASD group required a formal diagnosis of ASD by a licensed physician, along with official diagnostic documentation. Individuals with intellectual disabilities, severe psychiatric comorbidities, or major neurological conditions were excluded. TD participants reported no history of psychiatric, developmental, or neurological disorders. For all ASD participants, information regarding comorbid conditions, current medication use (if any), and intellectual functioning—measured via either intelligence quotient (IQ) or developmental quotient (DQ)—was obtained from the participants themselves or their legal guardians. All ASD participants (n = 18; 9 males, 9 females) provided written informed consent, with additional parental consent obtained for 17 of them. Prior to participation, all individuals completed the Japanese version of the Autism Spectrum Quotient (AQ) for adults.

Table 1 summarizes the clinical, cognitive, and diagnostic characteristics of participants in the ASD group, including comorbidities, medication status (noted as [+] for current medication, [−] for no medication), IQ or DQ scores, and AQ scores. Comorbid conditions such as attention deficit/hyperactivity disorder (ADHD), specific learning disorder (SLD), epilepsy, and color vision deficiency are indicated on an individual basis. For some participants, IQ or DQ values were not explicitly reported but confirmed to be within the normal range (WNR). Missing or unavailable information is marked with an em dash (“—”). These data were used to evaluate individual differences within the ASD group and to explore potential associations between participant profiles and synchrony performance.

This study was approved by the Ethics Committee of Kyoto University of Education (Approval No. 1805) and conducted in accordance with the Declaration of Helsinki. Written informed consent was obtained from all participants after they were provided with a comprehensive explanation of the study’s objectives, procedures, and data confidentiality measures. For ASD participants, verbal and written instructions were provided with time allocated for clarification.

### 2.2. Experimental Environment and Apparatus

Figure 1 illustrates the experimental setting. Each participant pair sat face-to-face at a table spaced 1.8 m apart, without wearing face coverings or headwear. The room temperature was maintained at an average of 20.7 °C, illumination was 1432 lx (CANA-0010, Tokyo Photoelectric Co., Tokyo, Japan), and ambient noise was 31.4 dB (CHE-SD1, Sanwa Supply, Okayama, Japan).

Body motion synchrony was measured using three wireless, three-axis accelerometers. Two sensors (TSND121, ATR-Promotions, Kyoto, Japan) were attached to each participant’s forehead with hypoallergenic adhesive tape. A third sensor (TSND151, ATR-Promotions, Japan) served as a synchronization marker by recording timestamped signals at the start and end of each experiment. Each device (37 mm × 46 mm × 12 mm; weight 22 g) transmitted time-series data via Bluetooth to a personal computer (Inspiron 15 7000, Dell Inc., Round Rock, TX, USA) at a sampling rate of 100 Hz. Data quality was monitored in real time to prevent Bluetooth dropouts, and experiments with >5% missing data were excluded. A calibration of the accelerometers was performed prior to each experiment using the manufacturer’s protocol.

To capture behavioral cues, a video camera (HDC-TM45, Panasonic, Osaka, Japan) recorded both participants during the experiment. These recordings were used only to verify behavioral compliance and were not subjected to further analysis. Video data were securely stored and anonymized.

### 2.3. Experimental Procedures

Participants engaged in a structured, face-to-face communication task designed to elicit naturalistic head movements under ecologically valid but experimentally controlled conditions. Each interaction involved one participant assuming the role of speaker and the other the role of listener. The communication took place in a quiet room with stable lighting and minimal environmental distraction. Participants were seated across from each other at a distance of 1.8 m and instructed to maintain a natural posture and gaze throughout the task. The primary goal was to simulate a realistic monologue-type conversation, allowing for the emergence of implicit nonverbal synchrony through nodding, subtle facial gestures, and other spontaneous body movements.

#### 2.3.1. TD Condition

In the TD condition, each dyad was composed of two TD participants who were randomly assigned to the roles of speaker and listener. The speaker received a printed article titled “Cashless Society”, adapted from the Japanese version of Wikipedia. This article, approximately 2600 characters in length, was selected because it covered a contemporary but neutral topic, ensuring low emotional valence and limited prior familiarity. Speakers were given time to read and mentally prepare their explanation before the task began, but they were not allowed to take notes or rehearse aloud.

During the interaction, the speaker was asked to explain the content of the article in their own words, maintaining a natural speaking style and steady eye contact. They were encouraged to avoid reading verbatim or using exaggerated gestures. The listener was instructed not to interrupt, ask questions, or initiate conversation.

However, to simulate genuine listening behavior, they were encouraged to exhibit backchannel responses such as nodding, brief affirmative sounds (e.g., “un”, “hai”, “ee”), and facial expressions conveying attention or understanding [35,36,37]. Both participants were instructed not to touch the head-mounted accelerometers or adjust their seating during the task.

To mark the beginning of the recording, the experimenter stood at the edge of the room and said, “The experiment is starting”, followed by a single clap, which served as a timestamp on the synchronization sensor. The experimenter then exited the room to minimize potential influence on the participants’ behavior. Upon completing the explanation, the speaker rang a small bell positioned on the table. The experimenter re-entered the room and concluded the task with the statement, “The experiment is ending”, followed by another clap to mark the end of data collection. These claps were synchronized with the third accelerometer and video recording to delineate the active recording window.

After the experiment, participants completed a brief questionnaire to assess subjective concentration and comprehension that included the following items:

Q1: “Were you able to concentrate on the explanation?”

Q2: “Was the content of the explanation easy to understand?”

Each item was rated on a five-point Likert scale ranging from 1 (strongly disagree) to 5 (strongly agree).

A trained researcher monitored the participants for signs of fatigue or discomfort and provided clarification if needed throughout the entire experiment. The task was continuously monitored via live audio and video feed from an adjacent control room to ensure participant safety without disrupting the natural flow of interaction.

#### 2.3.2. ASD Condition

In the ASD condition, each dyad was composed of one TD participant and one participant with ASD. The TD participant always assumed the role of speaker, while the ASD participant served as the listener. This fixed-role design was intentional and aimed to isolate the receptive aspects of nonverbal synchrony—particularly how individuals with ASD responded to the timing and rhythm of the speaker’s head movements and speech patterns.

The procedure was otherwise identical to the TD condition. The TD speaker received the same article (“Cashless Society”) and followed the same instructions for preparing and delivering their explanation. The ASD listener was given clear but simple instructions using visual aids and verbal clarification when necessary, ensuring full comprehension of the task expectations. In both conditions, the experimental procedures and environmental settings were held constant to ensure consistency across interactions.

Importantly, to minimize potential expectancy effects or social bias, the speakers were not informed of the diagnostic status of their communication partners. All participants were told only that they would be paired with another person for a communication task. There was no mention of clinical background or group membership. As a result, speakers in both the TD and ASD conditions were effectively blind to the listener’s diagnostic status, helping to ensure that their verbal and nonverbal behaviors were not influenced by assumptions or stereotypes.

This task structure allowed for the emergence of natural head motion synchrony, while maintaining control over the content and direction of communication. It also ensured consistency across dyads, enabling direct comparisons between groups based on movement patterns alone.

## 3. Data Analysis

### 3.1. Phase Difference Detection Algorithm

To detect body motion synchrony, we employed a phase difference detection algorithm previously proposed by Kwon et al. [30,31], which identifies temporal coordination in head movements based on acceleration signals (see Figure 2). The raw tri-axial (*X*, *Y*, *Z*) acceleration data, corresponding to sagittal, coronal, and transverse planes of head motion, were collected at a sampling frequency of 100 Hz (10 ms intervals).

The phase difference detection procedure employed in this study is described below. The magnitude of the acceleration vector was computed as the Euclidean norm:(1)$$a\left(t_{i}\right)=\sqrt{{a_{x}}^{2}\left(t_{i}\right)+{a_{y}}^{2}\left(t_{i}\right)+{a_{z}}^{2}\left(t_{i}\right)}\;\mathrm{for}\;i=0,\;1,\;2\;\ldots .$$

To eliminate individual differences in head movement amplitude caused by variation in speaking style or head size, normalization was performed using the z-score method:(2)$$a^{\prime }\left(t_{i}\right)=\frac{a\left(t_{i}\right)-\bar{a}}{\sigma_{a}}.$$

Here, $\bar{a}$ and $\sigma_{a}$ are calculated as(3)$$\bar{a}=\sum_{t_{i}\in T}\frac{a\left(t_{i}\right)}{\left|T\right|},$$(4)$$\sigma_{a}=\sqrt{\frac{\sum_{t_{i}\in T}{\left(\bar{a}-a\left(t_{i}\right)\right)}^{2}}{\left|T\right|-1}},$$
where $\bar{a}$ and $\sigma_{a}$ denote the mean and standard deviation of the acceleration over the entire recording duration, respectively.

To reduce high-frequency noise, the normalized signal was smoothed using a moving average over a 100 ms window:(5)$$\bar{a^{\prime }}\left(t_{i}\right)=\frac{1}{11}\sum_{l=i}^{i+10}a^{\prime }\left(t_{l}\right)\;\;\;\;\mathrm{for}\;i=0,\;1,\;2\;\ldots .$$

The smoothed signal was then used to detect local maxima (peaks), which indicate rhythmic head movements such as nodding or speaking-related oscillations. A sample was considered a peak if it satisfied the following condition:(6)$$\bar{a^{\prime }}\left(t_{i}\right)-\bar{a^{\prime }}\left(t_{i\pm 1}\right)>0.$$

To ensure that only meaningful movements were captured (i.e., strong enough to reflect communicative behavior), a peak amplitude threshold of 2.0 was applied, as established in previous studies [30,31].(7)$$\bar{a^{\prime }}\left(t_{i}\right)-2.0\geq 0.$$

For each pair of participants, the phase difference was defined as the smallest temporal offset between a peak from participant A and the nearest peak from participant B within a ±1.0 s window:(8)$$-1.0s\leq t_{j}-t_{i}\leq 1.0s.$$

Here, *t_i_* and *t_j_* denote the timestamps of the detected peaks for participant A and B, respectively. The resulting set of phase differences formed the basis for synchrony analysis.

### 3.2. Synchrony Feature Extraction

Using the distribution of detected phase differences over the course of each interaction, four statistical features were extracted to quantitatively characterize head motion synchrony:Density (Synchrony Activity): The number of synchronized events per minute, reflecting the overall activity level of synchrony within a dyad.Mean Phase Difference (Synchrony Directionality): The average temporal lead or lag between paired movements. Positive values indicate that the speaker’s movements consistently preceded those of the listener, while negative values indicate the opposite. This measure captures the directional dynamics of synchrony and reflects potential leader–follower roles within the interaction.Standard Deviation (Synchrony Variability): The dispersion of phase differences around the mean. Smaller values indicate more temporally precise alignment, whereas larger values reflect greater inconsistency in coordination.Kurtosis (Synchrony Coherence): A measure of the peakedness of the phase difference distribution. Higher kurtosis indicates stronger convergence of synchronized movements around the mean phase, implying greater coherence and temporal stability.

Together, these four features—activity, directionality, variability, and coherence—offer a multidimensional representation of body motion synchrony during interpersonal communication.

## 4. Results

### 4.1. Phase Difference Distributions

To assess the temporal characteristics of head motion synchrony, we examined the distribution of phase differences between speaker and listener head movements in both the TD and ASD groups. Figure 3 illustrates the normalized distributions of head motion phase differences for the TD and ASD groups, aggregated across all dyads. The *x*-axis represents the temporal offset in milliseconds between the peak head movements of paired participants (with positive values indicating that the listener lagged behind the speaker), while the *y*-axis indicates the relative frequency within 100 ms bins.

In the TD group, approximately 67.4% of all synchrony events occurred within the ±300 ms window centered around the mean phase difference, indicating tightly clustered, highly coordinated, and temporally precise alignment. The distribution was sharply peaked and symmetrically centered near the mean, consistent with high temporal coherence and rapid mutual entrainment. The phase difference distribution showed a rapid decline on either side of the peak, suggesting that most synchrony events occurred with minimal temporal delay.

In contrast, the ASD group exhibited a significantly flatter and more dispersed distribution, with only 41.7% of events falling within the ±300 ms range. The distribution showed long tails extending beyond ±500 ms, with a particularly extended tail toward negative lags. Since negative lags reflect listener movements that precede the speaker’s behavior, this may indicate inconsistent listener responses. In addition, the histogram revealed irregular, non-uniform fluctuations across the entire distribution, including an unexpected secondary rise near –1000 ms, which may reflect atypical anticipatory movements or misaligned rhythmic timing. These patterns suggest that head motion synchrony in the ASD group not only lacks precision but also exhibits unstable entrainment dynamics. Collectively, these findings reflect lower synchrony precision and weaker temporal convergence in the ASD group, consistent with the increased standard deviation and reduced kurtosis observed in the statistical analysis.

Together, these findings support the interpretation that TD–TD pairs exhibit stronger and more temporally precise synchrony, while TD–ASD pairs show weaker and temporally diffuse coordination. These distributional patterns are consistent with the statistical results reported in the subsequent sections for standard deviation and kurtosis.

### 4.2. Synchrony Metrics Across Groups

To characterize body motion synchrony, we employed four quantitative indices derived from the phase difference distributions: synchrony activity, directionality, variability, and coherence. Figure 4 shows raincloud plots, illustrating the group-level differences between TD and ASD participants across four synchrony-related metrics. Each violin plot shows the distribution of data, the boxplots show the median and interquartile range, and individual observations are marked with scatter points. Detailed pair-level data for each group are provided in the Appendix A.

#### 4.2.1. Synchrony Activity

The mean relative frequency was 9.63 times/min (*SD* = 3.23) for the TD group and 5.96 times/min (*SD* = 2.73) for the ASD group. An independent samples *t*-test revealed a statistically significant difference in synchrony activity between groups—*t*(34) = 3.68, *p* = 0.0008, Cohen’s *d* = 1.23—indicating a large effect size. ASD participants exhibited less frequent synchronized head movements compared to TD participants.

#### 4.2.2. Synchrony Directionality

The mean phase difference was 90.0 ms (*SD* = 61.6) for the TD group and 84.4 ms (*SD* = 187.2) for the ASD group. Although both groups exhibited positive phase differences, suggesting listener-delayed responses, the ASD group showed considerably larger variance and more outliers. However, the difference between groups was not statistically significant—*t*(34) = 0.12, *p* = 0.91, Cohen’s *d* = 0.04.

#### 4.2.3. Synchrony Variability

The TD group exhibited a mean standard deviation of 311.1 ms (*SD* = 51.8), while the ASD group showed a significantly greater mean of 384.4 ms (*SD* = 86.1). This difference was statistically significant—*t*(34) = –3.10, *p* = 0.0039, Cohen’s *d* = –1.03—indicating a large effect size, with TD–ASD dyads showing greater temporal variability than TD–TD dyads.

#### 4.2.4. Synchrony Coherence

Kurtosis, which reflects the convergence of the phase difference distribution toward the mean, was higher in the TD group (*M* = 0.81, *SD* = 0.84) than in the ASD group (*M* = 0.13, *SD* = 1.01). This difference was statistically significant—*t*(34) = 2.20, *p* = 0.036, Cohen’s *d* = 0.73—suggesting weaker temporal convergence and reduced synchrony coherence in ASD interactions.

### 4.3. Control Analyses for Potential Confounds

#### 4.3.1. Autism Spectrum Quotient (AQ) Comparisons

To examine whether individual differences in autistic traits could account for group differences in head motion synchrony, we analyzed the Autism Spectrum Quotient (AQ) scores for all participants. First, we compared the AQ scores of TD speakers assigned to the TD condition and those assigned to the ASD condition. The mean AQ scores were 16.56 (*SD* = 6.17) for the TD condition and 15.94 (*SD* = 8.36) for the ASD condition. An independent samples *t*-test indicated no significant difference between these groups (*t*(34) = 0.25, *p* = 0.804), suggesting that the TD speakers across conditions were similar in their autistic trait profiles.

In contrast, ASD listeners showed substantially higher AQ scores (*M* = 30.56, *SD* = 7.47) than TD listeners (*M* = 15.56, *SD* = 6.21). This difference was statistically significant (*t*(34) = –6.55, *p* < 0.0001), reflecting the diagnostic distinction between groups. These findings confirm that AQ traits among TD participants were balanced across conditions, and that group-level differences in synchrony are unlikely to be driven by AQ variation in the TD population. Full AQ data are provided in Appendix A.

#### 4.3.2. TD Speaker Behavior Across Conditions

To examine whether TD speakers behaved differently depending on the diagnostic status of their communication partner, we compared the density of head movements in the TD and ASD conditions. The mean head movement density was 23.30 nods per minute (*SD* = 4.83) for TD speakers in the TD condition and 24.27 nods per minute (*SD* = 5.17) in the ASD condition. An independent samples *t*-test revealed no significant difference between conditions—*t*(34) = –0.56, *p* = 0.576, Cohen’s *d* = –0.19.

These results suggest that TD speakers exhibited comparable nonverbal behavior across both conditions, supporting the interpretation that the reduced synchrony observed in TD–ASD dyads is primarily attributable to characteristics of the ASD participants rather than differences in TD speaker behavior. Pairwise statistics are provided in Appendix A.

#### 4.3.3. Listener Engagement and Comprehension

To address whether differences in listener engagement may account for the observed group-level differences in synchrony, we compared post-task self-reported ratings of concentration and comprehension between TD and ASD listeners. Each item was rated on a five-point Likert scale ranging from 1 (strongly disagree) to 5 (strongly agree). Concentration ratings were similar between groups, with TD listeners reporting a mean score of 3.28 (*SD* = 1.32) and ASD listeners 3.44 (*SD* = 1.29). Comprehension scores were also comparable, with means of 3.44 (*SD* = 1.15) for TD listeners and 3.94 (*SD* = 1.16) for ASD listeners. Mann–Whitney U tests indicated no statistically significant differences for either concentration (*U* = 173.5, *p* = 0.720) or comprehension (*U* = 206.0, *p* = 0.149). These findings suggest that listener-level differences in self-perceived engagement or understanding do not account for the reduced synchrony observed in TD–ASD dyads.

#### 4.3.4. Influence of Experiment Duration on Synchrony Measures

To examine the potential effect of experiment duration on synchrony, we conducted Pearson correlation analyses between measurement period and each synchrony feature (density, mean phase difference, SD, and kurtosis). Using the combined data from the TD and ASD groups, no significant correlations were observed: density (*r* = –0.0004, *p* = 0.998), mean phase difference (*r* = –0.05, *p* = 0.759), SD (*r* = –0.03, *p* = 0.840), and kurtosis (*r* = 0.12, *p* = 0.452). These results indicate that experiment duration did not affect the synchrony measures.

## 5. Discussion

This study provides novel insights into the temporal dynamics of nonverbal synchrony in face-to-face communication between typically developing (TD) individuals and individuals with autism spectrum disorder (ASD). Using a phase difference detection algorithm and four statistical measures—relative frequency, mean phase difference, standard deviation, and kurtosis—we have systematically quantified synchrony not only in terms of frequency but also in terms of its temporal consistency and convergence strength. Our results demonstrate clear group-level differences, particularly in the dispersion and stability of synchrony patterns, shedding light on the sensorimotor basis of social interaction in ASD. This study advances the field by refining methodological approaches to measuring synchrony and by identifying novel markers of impaired coordination that can contribute to both theoretical models of autism and clinical applications.

### 5.1. Methodological Advancements and Novel Contributions to Synchrony Research

A key contribution of this study lies in its methodological innovation using a phase difference detection algorithm, which enables the high-resolution, time-continuous quantification of nonverbal synchrony based on triaxial acceleration data. Unlike traditional approaches that rely on cross-correlation [25] or discrete event counts of co-occurring behaviors [17], this algorithm captures the instantaneous phase relationship between dyadic movements, offering a finer-grained analysis of temporal coordination. By computing phase differences at each moment in time and constructing full distributions, the method provides dynamic, directional, and data-rich insights into interpersonal alignment. This level of temporal precision is particularly advantageous in studying ASD, where subtle disruptions in synchrony may occur even in the absence of gross behavioral abnormalities.

Furthermore, the integration of distribution-based statistical features enhances the utility of the phase difference detection algorithm by enabling the multidimensional characterization of synchrony patterns. Relative frequency quantifies the overall activity level of synchrony throughout the experimental interaction, while mean phase difference captures the directional lead–lag structure between speaker and listener. Standard deviation reflects temporal variability, and kurtosis indicates the convergence strength of synchronization events around the central phase. Together, these features allow researchers to move beyond binary classifications of synchrony presence or absence, and instead to evaluate the richness, consistency, and directionality of coordination over time. This level of granularity is particularly valuable for identifying atypical social timing profiles in ASD populations, and for exploring individual differences that may relate to cognitive, sensory, or affective traits.

Additionally, this study uniquely focuses on a controlled unidirectional communication context, in which ASD participants exclusively assumed the listener role. This design isolates the receptive dimension of social timing and allows for a more precise assessment of implicit synchrony processes. Many prior studies on ASD-related synchrony deficits have focused on interactive or reciprocal paradigms [22,24,38,39], where compensatory mechanisms—such as explicit effortful imitation—may obscure underlying impairments in automatic timing mechanisms. By eliminating bidirectional conversational demands, our study provides a cleaner test of nonverbal entrainment deficits.

### 5.2. Synchrony Activity and Temporal Precision: Quantitative and Qualitative Divergences

Consistent with prior research emphasizing the role of sensorimotor coupling in social bonding [11,12,13,14,40,41], we found that TD–TD pairs exhibited significantly higher relative frequencies of synchronized head movements compared to TD–ASD pairs. This suggests that TD individuals engage in spontaneous, fine-grained sensorimotor alignment more frequently, even in the absence of explicit verbal cues and during structured, one-way verbal communication. Such synchronization has been considered an implicit social mechanism for enhancing rapport and shared understanding [13,15,16,40]. It has also been linked to affiliation, social cohesion, and shared attention, reflecting both the interpersonal quality of dyadic interactions and the broader social bonding functions of synchrony [12,14,18,42,43,44].

In contrast, the ASD group demonstrated a significantly reduced frequency of synchronized movements. These findings expand upon previous reports of attenuated social motor synchronization in individuals with ASD [22,25] and suggest that even in the absence of explicit conversational turn-taking, implicit nonverbal synchrony is diminished. These outcomes align with findings by Koehler et al. (2022), who reported reduced synchrony in adults with ASD across various social dyad types, especially in head and upper body motion [26].

Additionally, the ASD group exhibited significantly greater variability in phase differences, as reflected in increased standard deviation and reduced kurtosis. These results indicate that even when synchrony occurs, it is more temporally dispersed and lacks the convergence observed in TD–TD dyads. Notably, the presence of extreme phase lags (exceeding ±500 ms) suggests that while ASD individuals can align their head movements with their communication partners, the reliability and precision of that alignment are impaired. This pattern is consistent with findings from Isaksson et al. (2018) and Oberman et al. (2009), who reported timing irregularities in motor and social behaviors among individuals with ASD [24,45]. Given the growing evidence that ASD-related social difficulties may stem, at least in part, from predictive coding deficits [46,47], our findings further support the idea that disruptions in temporal coordination may underlie broader social impairments.

In addition to reduced temporal precision, emerging research highlights that individuals with ASD may express different vitality forms—that is, the dynamic and expressive manner in which actions are performed [48]. Casartelli et al. (2020) demonstrated that neurotypical adults often struggle to interpret the vitality forms of children with ASD, suggesting that interpersonal misattunement may arise from perceptual asymmetries rather than from timing deficits alone [49]. Therefore, diminished synchrony may reflect not only individual impairments but also mismatched expressive styles between interlocutors. To disentangle synchrony impairments from mismatches in communicative style, future research should systematically manipulate speaker–listener pairings, including ASD speakers and TD listeners, and analyze the perceptual reciprocity of expressive motor dynamics.

### 5.3. Phase Distributions and Entrainment Signatures

A key insight from this study comes from the distributional analysis of phase differences. TD–TD dyads displayed tightly clustered, symmetrical phase distributions centered near zero, whereas TD–ASD dyads exhibited flatter and broader distributions with long tails—particularly toward positive lags. These patterns suggest a breakdown in fine-grained interpersonal entrainment [50,51]. In neurotypical interactions, phase distributions are expected to exhibit a sharp peak near zero, reflecting robust and temporally stable coordination. The flatter distributions observed in TD–ASD dyads indicate that synchrony, when present, is highly inconsistent and less precisely timed.

This divergence aligns with research on neural and behavioral entrainment in ASD, which has demonstrated weaker phase-locking to external stimuli [52,53,54]. Importantly, disrupted synchrony is not merely a consequence of general motor impairments; rather, it appears to be specifically linked to difficulties in dynamically adjusting one’s behavior to social cues in real time [55].

### 5.4. Trait-Level Explanations for Reduced Synchrony in ASD: The Roles of AQ and IQ

The current analysis of Autism Spectrum Quotient (AQ) scores helps clarify the role of individual differences in autistic traits in shaping group-level differences in head motion synchrony. Our findings show that TD speakers assigned to the TD and ASD conditions did not differ significantly in their AQ scores, indicating that autistic trait levels were balanced across the two experimental groups. In addition, to examine whether TD speakers modulated their behavior based on the diagnostic status of their communication partner, we compared head movement density between the TD and ASD conditions. This analysis revealed no significant differences between conditions, suggesting that TD speakers exhibited consistent levels of nonverbal behavior, regardless of their partner’s diagnostic background. These findings support the interpretation that the reduced synchrony observed in TD–ASD dyads cannot be attributed to variation in autistic traits or behavioral modulation among TD participants.

As expected, AQ scores were significantly higher among ASD listeners than TD listeners, consistent with their diagnostic status. This robust group difference confirms the integrity of the diagnostic contrast and suggests that elevated autistic traits in the ASD group are a core component of the observed reduction in nonverbal synchrony. Taken together, these findings strengthen the argument that reduced head motion synchrony in ASD is not an artifact of sample-level trait heterogeneity, but rather reflects a meaningful social-communicative difference characteristic of autism. Future studies may benefit from examining how specific AQ subdomains—such as attention to detail or social skills—relate to different dimensions of nonverbal synchrony.

One limitation of the present study is that IQ data were available only for the ASD group, and thus the two groups were not explicitly matched for intellectual functioning. Previous studies have shown that even in samples where ASD and TD groups were matched on IQ, or where all participants had average or above-average intelligence, individuals with ASD consistently exhibited impairments in nonverbal synchrony and social coordination [22,25]. These findings support the interpretation that synchrony deficits in ASD reflect a core feature of the condition rather than being secondary to general cognitive delay. However, other research has shown that motor performance is positively correlated with IQ in individuals with ASD [56]. In particular, children with lower IQ exhibited poorer gross motor skills and made more mirroring errors compared to those with higher IQ levels [56]. These findings suggest that IQ may modulate the severity of motor coordination difficulties in ASD. Therefore, future research would benefit from including comprehensive cognitive assessments in both groups to better isolate the respective contributions of intellectual functioning and autism-specific impairments in nonverbal coordination.

### 5.5. Consideration of Outliers in Synchrony Data

Although one dyad (Pair ID 9) in the ASD group exhibited extreme values in certain synchrony measures—particularly mean phase difference, standard deviation, and kurtosis—the exclusion of this data point did not alter the overall pattern of results (see Appendix A). Specifically, statistical group differences in synchrony activity and temporal variability remained significant, suggesting that the findings were robust and not driven by a single outlier (see Appendix A).

Importantly, a behavioral review of the recorded interaction revealed that both participants in this dyad remained attentive and engaged throughout the task. This observation supported the inclusion of the data, as the atypical synchrony metrics likely reflected genuine individual differences rather than technical artifacts or protocol violations [22,25].

Future studies should further examine the implications of individual differences in synchrony metrics, including those that fall at the statistical extremes. Rather than treating such data as noise, it may be fruitful to explore how atypical synchrony patterns relate to specific cognitive, behavioral, or diagnostic subtypes within the autism spectrum.

Additionally, robust statistical frameworks—such as median-based or nonparametric approaches—may be applied to complement mean-based comparisons, ensuring that the analysis remains sensitive while minimizing the influence of extreme values. Combining behavioral validation with robust statistics will be crucial for improving the interpretability and generalizability of synchrony-based metrics in clinical research.

### 5.6. Implications for ASD Assessment and Intervention

The synchrony metrics used in this study—particularly standard deviation and kurtosis—offer promising new tools for ASD assessment. Unlike traditional behavioral coding, which relies on subjective interpretation, these measures provide continuous, quantifiable, and nonverbal indices of dyadic coordination. Such metrics could be particularly valuable for individuals with limited expressive language, as they do not require explicit verbal responses.

From a clinical perspective, synchrony-based interventions may be beneficial for individuals with ASD. Sensorimotor training programs, virtual reality social simulations, and biofeedback paradigms could leverage synchrony metrics to provide real-time feedback and improve social coordination skills. Recent studies have shown that targeted motor and rhythm training can enhance social responsiveness in ASD [23,57,58,59,60], suggesting that synchronization-based therapies may hold promise for intervention.

Furthermore, these findings align with dynamical systems models of social interaction, which emphasize continuous real-time coupling between agents [55]. Understanding ASD-related impairments in social timing from this perspective shifts the focus away from static social “deficits” and toward more dynamic, context-dependent explanations.

### 5.7. Limitations and Future Directions

Despite its contributions, this study has several limitations. First, while our use of unidirectional verbal communication allowed for controlled comparisons, it does not fully capture the complexities of natural social interactions. Future studies should examine synchrony in both dyadic and multi-party conversations involving reciprocal turn-taking to provide a more holistic understanding of interpersonal coordination. Furthermore, incorporating multimodal data encompassing multisensory input, nonverbal behavior, and physiological signals may enhance the analytical framework for investigating complex social dynamics [61,62,63,64,65].

Second, in contrast to traditional neuroimaging or neurophysiological approaches, the present method is based on non-invasive, portable, and relatively low-cost technologies such as accelerometers and video cameras. This simplicity enhances its feasibility for implementation in clinical environments with limited resources. However, systematic cost–benefit evaluations and pilot implementations in real-world clinical settings are warranted to assess feasibility across diverse populations. To expand its utility, future studies should explore how this framework can be practically adapted and scaled for use in diverse real-world clinical settings, particularly to improve accessibility for children with limited healthcare access. Specifically, we should aim to adapt and field test this framework in community-based clinical contexts, such as public health centers, school-based screening programs, or mobile outreach services.

Additionally, simplified sensor setups (e.g., single-sensor configurations) and automated analysis pipelines could be developed to further reduce cost and technical barriers to adoption. In this regard, ongoing work is already investigating the use of camera-based systems for the real-time detection and visualization of head motion synchrony, with the goal of eliminating the need for wearable sensors and facilitating seamless integration into clinical settings. This direction is outlined in our previous study, which introduces a practical visualization system for real-time synchrony detection using an RGB video input [66]. Such innovations are expected to enhance the accessibility and scalability of synchrony-based screening tools in everyday healthcare environments.

## 6. Conclusions

This study provides compelling evidence that individuals with ASD exhibit significant impairments in nonverbal synchrony, particularly in the temporal precision and stability of interpersonal coordination. By leveraging a phase difference detection algorithm and distribution-based features, we offer a robust and high-resolution framework for quantifying synchrony and identifying its disruptions in ASD. These synchrony metrics—continuous, nonverbal, and language-independent—show strong potential as behavioral biomarkers for autism, supporting scalable, objective tools for clinical assessment. Our findings contribute to the growing literature emphasizing social timing as a core component of autism-related difficulties and highlight the clinical utility of synchrony-based approaches in both research and practice.

## Figures and Tables

**Figure 1 diagnostics-15-01268-f001:**
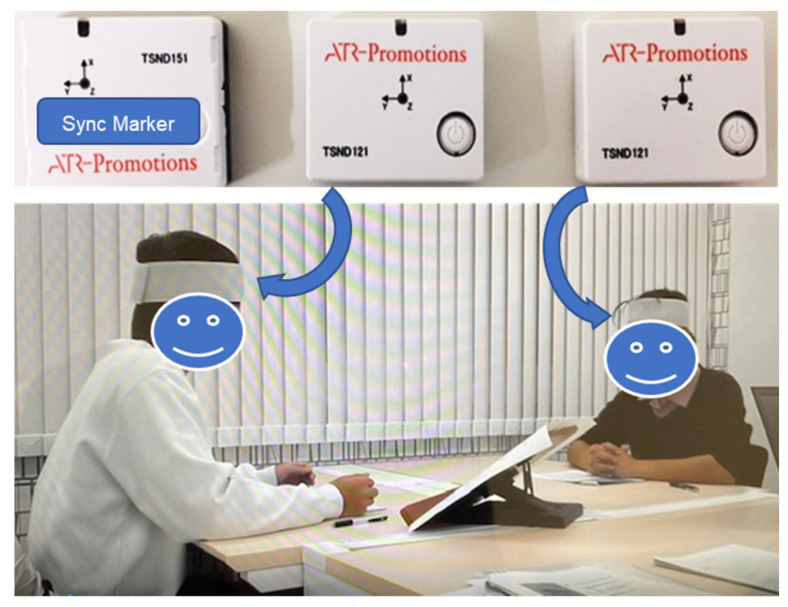
Experimental setup and placement of the acceleration sensors.

**Figure 2 diagnostics-15-01268-f002:**
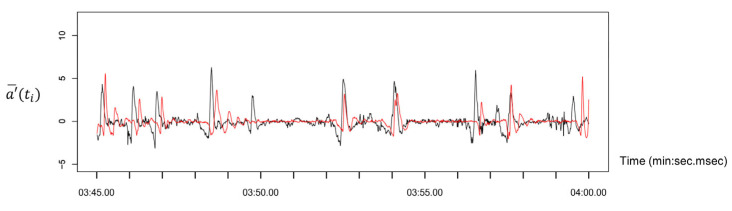
Representative time-series data of head movement during the unidirectional verbal communication task [31]. The black line represents the speaker’s head acceleration, while the red line depicts the listener’s head acceleration.

**Figure 3 diagnostics-15-01268-f003:**
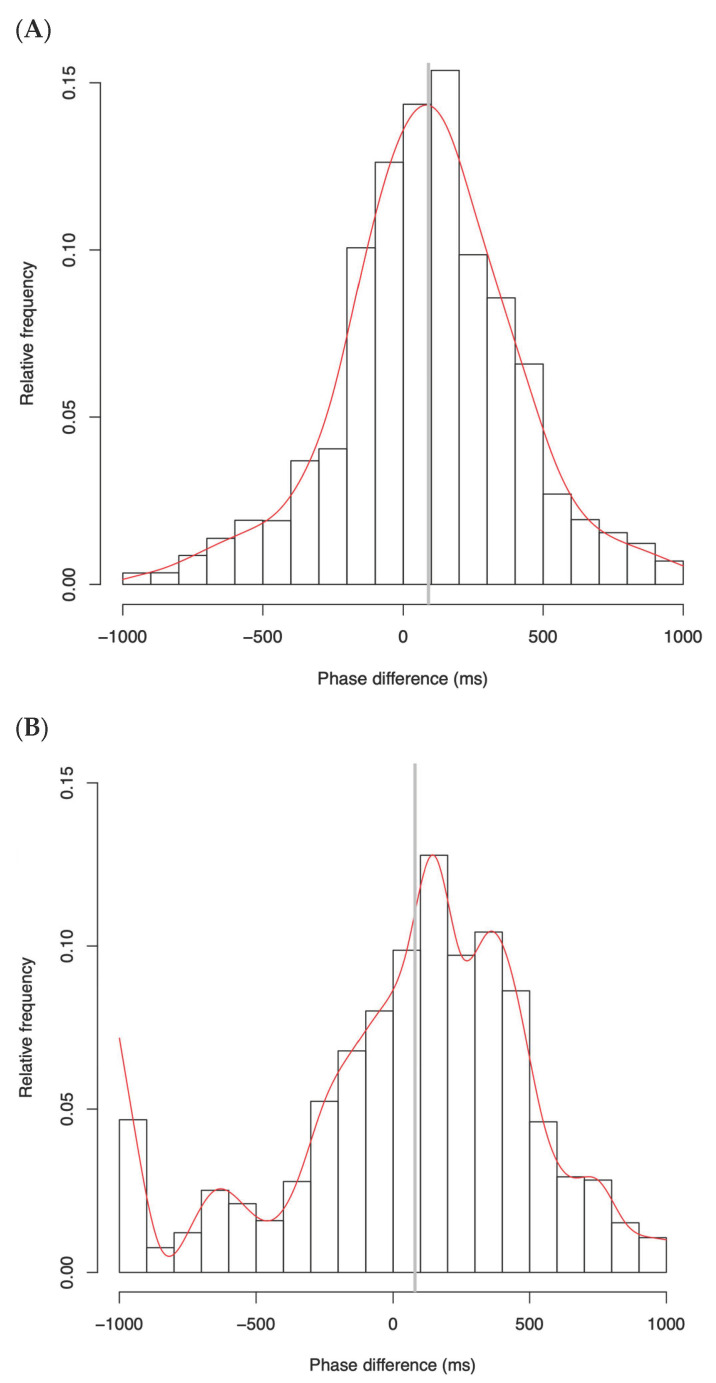
Phase difference distributions of head movements across all dyads in the TD group (**A**) and the ASD group (**B**). The horizontal axis denotes the phase difference between speaker and listener head movements (in milliseconds), where positive values indicate that the listener’s movement lags behind the speaker’s, and negative values indicate the reverse. The vertical axis represents the relative frequency of phase difference occurrences, binned in 100 ms intervals. The gray vertical line marks the overall mean of the distribution, and the red curve represents a smoothing spline fitted to the data. The TD group displays a sharply peaked, symmetrical distribution centered near the mean, reflecting high temporal precision in interpersonal synchrony. In contrast, the ASD group exhibits a flatter, more dispersed distribution, indicative of reduced synchrony precision and greater temporal variability.

**Figure 4 diagnostics-15-01268-f004:**
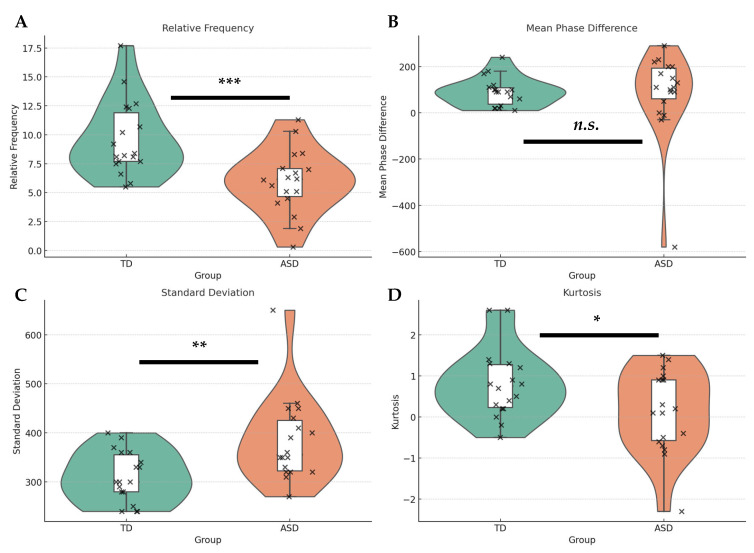
Raincloud plots illustrate group-level differences between TD and ASD participants across four synchrony-related metrics. Each plot combines a violin plot (distribution), a boxplot (median and interquartile range), and individual data points. The panels show the following: (**A**) relative frequency, (**B**) mean phase difference, (**C**) standard deviation, and (**D**) kurtosis. Asterisks indicate statistically significant differences (* *p* < 0.05, ** *p* < 0.01, *** *p* < 0.001); “*n.s.*” denotes non-significance.

**Table 1 diagnostics-15-01268-t001:** Clinical, cognitive, and diagnostic characteristics of ASD participants.

ASD Participant No. (Gender)	Comorbidities and Medication (+/−)	IQ/DQ	AQ
A1 (F)	ADHD (−)	92	30
A2 (F)	—	Average Range	28
A3 (F)	—	Average Range	30
A4 (F)	—	97	21
A5 (M)	Epilepsy (+)	87	38
A6 (M)	—	121	33
A7 (M)	—	102	35
A8 (M)	—	75	34
A9 (F)	ADHD (+)	87	38
A10 (M)	ADHD (−)	115	36
A11 (M)	SLD	86	35
A12 (F)	—	139	30
A13 (M)	—	86	10
A14 (M)	Color Vision Deficiency	93	32
A15 (M)	—	93	22
A16 (F)	SLD, ADHD (−)	111	37
A17 (F)	—	97	38
A18 (F)	—	69	23

## Data Availability

The original contributions presented in this study are included in this article; further inquiries can be directed to the corresponding author. Code availability: Made available on request.

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
