# Peer review of "Quantifying Body Motion Synchrony in Autism Spectrum Disorder Using a Phase Difference Detection Algorithm: Toward a Novel Behavioral Biomarker"

_diagnostics, 2025, doi:10.3390/diagnostics15101268_

Round 1

Reviewer 1 Report

Comments and Suggestions for Authors

The study conducted by Kwon and Kotani aims to assess whether and how individuals with ASD synchronize their head movements during face-to-face social communication. The idea of using continuous measures of phase difference to examine ecological social dynamics is interesting, and I agree that it has potential clinical implications. However, there are methodological issues that temper my enthusiasm.

The sampling and the grouping strategy are unclear, not well explained, and poorly justified.

In the method section, the authors stated: “A total of 72 participants took part in this study, divided equally into the typically developing (TD) group and the autism spectrum disorder (ASD) group” suggesting that there are 36 TD and 36 ASD.

Later, the dyad assignments are defined conditions “In the TD condition, speaker–listener roles were randomly assigned. In the ASD condition, the TD participant always served as the speaker, while the ASD participant was always assigned the listener role.” This could suggest that TD performed the TD condition and the ASD condition.

In line 197, it is reported that participants were randomly assigned as either speakers or listener, but this is not the case for ASD participants.

It should be at least better discussed the aspect that ASD individuals are only listeners, while the speakers are always TD. What would the results look like if individuals with ASD were the speakers and TD participants were the listeners? Does the diminished and inconsistent synchronization pattern characterize ASD individuals specifically, or is it rather a matter of motor style similarity between the speaker and the listener? See Casartelli et al., 2020 “Neurotypical individuals fail to understand action vitality form in children with autism spectrum disorder” as an example of this perspective.

It should be important to clarify that there are no differences between TD speakers in TD-TD dyads and TD-ASD dyads.

The TD and ASD groups are not reported to be IQ-matched, despite the fact that this would be important in order to rule out the possibility that a lower IQ alone could explain the anomalies in the synchrony metrics observed in the ASD group.

In the Participants section, the authors stated that clinical, cognitive, and diagnostic characteristics of participants in the ASD group were used to evaluate individual differences within the ASD group and to explore potential associations between participant profiles and synchrony performance. In the discussion section, it is raised the possibility that IQ could explain some variability, but there are no analyses. Please report analyses and clarify the discussed aspects by assessing separately the relationship between AQ and IQ with the synchronization profile.

The experimental procedure should be improved. Condition, trial, and session are not well reported. For example, in line 197, it is stated that “Prior to each trial, participants were randomly assigned as either speaker or listener.” However, it is not clear what a trial is, there seems to be only one trial in each experimental session. Please describe the procedure in more detail and be consistent with the names.

What about the concentration and comprehension questions? Is there any difference between TD and ASD listeners? Do they explain any variability in the listener synchronization?

Were the speakers aware if the listener was a TD or ASD? Did you control this aspect?

Data reported in Tables S1 and S2 highlight that there are much more variabilities in the session duration of the TD-ASD group than in the TD-TD group. Did you check the possible impact of different durations on synchronization measures?

Moreover, Pair 9 seems to be an outlier in the mean phase difference, the standard deviation, and the kurtosis. Did you check whether the data are stable without this outlier?

Author Response

Comment 1: The study conducted by Kwon and Kotani aims to assess whether and how individuals with ASD synchronize their head movements during face-to-face social communication. The idea of using continuous measures of phase difference to examine ecological social dynamics is interesting, and I agree that it has potential clinical implications. However, there are methodological issues that temper my enthusiasm.

Our Response 1: We sincerely thank the reviewer for their overall positive assessment of our study and for recognizing the conceptual contribution of using continuous, phase-based measures to capture ecological aspects of social interaction in ASD. We appreciate your acknowledgment of the potential clinical relevance of our findings.

At the same time, we understand and respect your concerns regarding methodological aspects of the study. In response, we have carefully addressed each of the specific methodological issues raised in your subsequent comments (Comments 2–10), and we have made substantial revisions throughout the manuscript to reflect and incorporate your suggestions.

We hope that these revisions address your concerns and contribute to strengthening the clarity, transparency, and methodological rigor of the manuscript.

Comment 2: The sampling and the grouping strategy are unclear, not well explained, and poorly justified. In the method section, the authors stated: “A total of 72 participants took part in this study, divided equally into the typically developing (TD) group and the autism spectrum disorder (ASD) group” suggesting that there are 36 TD and 36 ASD.

Later, the dyad assignments are defined conditions “In the TD condition, speaker–listener roles were randomly assigned. In the ASD condition, the TD participant always served as the speaker, while the ASD participant was always assigned the listener role.” This could suggest that TD performed the TD condition and the ASD condition.

In line 197, it is reported that participants were randomly assigned as either speakers or listener, but this is not the case for ASD participants.

Our Response 2: We appreciate the reviewer’s thoughtful feedback regarding the clarity of our sampling and grouping strategy. In the revised manuscript, we have clarified this point in the Participants section (Section 2.1).

Specifically, we now state that the study included 54 typically developing (TD) individuals and 18 individuals with autism spectrum disorder (ASD). Among the TD participants, 36 were assigned to the TD condition to form 18 TD–TD dyads, and the remaining 18 TD participants were assigned to the ASD condition, each paired with an ASD participant to form 18 TD–ASD dyads. This results in a total of 36 dyads, with each dyad consisting of two same-gender individuals who had no prior acquaintance. This clarification was made to prevent the misunderstanding that the same TD participants were involved in both conditions, which was not the case. Each participant took part in only one interaction.

Regarding the reviewer’s observation about line 197 and the random assignment of speaker and listener roles, we kindly refer to our Response to Comment 6, where we have addressed this point in detail and clarified the procedures for role assignment in both the TD and ASD conditions.

We hope this revision addresses the reviewer’s concerns and helps improve the methodological transparency of our study.

[Revised Sentences, Pages 3-4, Lines 128-151]

(Before)

2.1. Participants

A total of 72 participants took part in this study, divided equally into the typically developing (TD) group and the autism spectrum disorder (ASD) group. Each group comprised 36 participants (18 males and 18 females). The mean age of the TD group was 21 years (SD = 0.92), and that of the ASD group was 23 years (SD = 1.65). All participants were native Japanese speakers with normal or corrected-to-normal vision and normal hearing. To control for the potential effects of interpersonal familiarity on body motion synchrony [12, 14, 17], each dyad was composed of two same-gender individuals who had no prior acquaintance. Gender ratios were strictly balanced (1:1) across all conditions. In every dyad, one participant was assigned the role of speaker and the other the listener. In the TD condition, speaker–listener roles were randomly assigned. In the ASD condition, the TD participant always served as the speaker, while the ASD participant was always assigned the listener role.

Inclusion criteria for the ASD group required a formal diagnosis of ASD by a licensed physician, along with official diagnostic documentation. Individuals with intellectual disabilities, severe psychiatric comorbidities, or major neurological conditions were excluded. TD participants reported no history of psychiatric, developmental, or neurological disorders. For all participants, information regarding comorbid conditions, current medication use (if any), and intellectual functioning—measured via either intelligence quotient (IQ) or developmental quotient (DQ)—was obtained from participants themselves or their legal guardians. All ASD participants (n = 18; 9 males, 9 females) provided written informed consent, with additional parental consent obtained for 17 of them. Prior to participation, all individuals completed the Japanese version of the Autism-Spectrum Quotient (AQ) for adults.

(After)

2.1. Participants

A total of 72 participants took part in this study, including 54 typically developing (TD) individuals and 18 individuals with autism spectrum disorder (ASD). The 54 TD participants were assigned to two different conditions: 36 participated in the TD condition, forming 18 TD–TD dyads, and the remaining 18 were assigned to the ASD condition, where each TD participant was paired with an ASD participant to form 18 TD–ASD dyads. In total, 36 dyads were created for the experiment. These two groups of TD participants were mutually exclusive, and no individual participated in both conditions. All TD participants were undergraduate students at Kyoto University of Education. The mean age of the TD group was 21 years (SD = 0.92), and that of the ASD group was 23 years (SD = 1.65). All participants were native Japanese speakers with normal or corrected-to-normal vision and normal hearing. To control for the potential effects of interpersonal familiarity on body motion synchrony [12, 14, 17], each dyad was composed of two same-gender individuals who had no prior acquaintance. Gender ratios were strictly balanced (1:1) across all conditions.  

Inclusion criteria for the ASD group required a formal diagnosis of ASD by a li-censed physician, along with official diagnostic documentation. Individuals with intellectual disabilities, severe psychiatric comorbidities, or major neurological conditions were excluded. TD participants reported no history of psychiatric, developmental, or neurological disorders. For all participants, information regarding comorbid conditions, current medication use (if any), and intellectual functioning—measured via either intelligence quotient (IQ) or developmental quotient (DQ)—was obtained from participants themselves or their legal guardians. All ASD participants (n = 18; 9 males, 9 females) provided written informed consent, with additional parental consent obtained for 17 of them.

Comment 3: It should be at least better discussed the aspect that ASD individuals are only listeners, while the speakers are always TD. What would the results look like if individuals with ASD were the speakers and TD participants were the listeners? Does the diminished and inconsistent synchronization pattern characterize ASD individuals specifically, or is it rather a matter of motor style similarity between the speaker and the listener? See Casartelli et al., 2020 “Neurotypical individuals fail to understand action vitality form in children with autism spectrum disorder” as an example of this perspective.

Our Response 3: Thank you for the insightful comment. We agree that using only ASD listeners limits the interpretation of synchrony disruptions. As you pointed out, it is important to note that disrupted synchrony may not be solely due to ASD-specific timing deficits, but also to mismatched action vitality forms between TD speakers and ASD listeners. Casartelli et al. (2020) demonstrated that neurotypical adults often fail to interpret the vitality forms expressed by children with ASD, suggesting a possible asymmetry in perception–action coupling that may contribute to reduced social attunement, even when timing abilities are preserved. We have now added this discussion to the revised manuscript.

[Added Sentences, Page 12, Lines 462-472]

In addition to reduced temporal precision, emerging research highlights that individuals with ASD may express different vitality forms—that is, the dynamic and expressive manner in which actions are performed [48]. Casartelli et al. (2020) demonstrated that neurotypical adults often struggle to interpret the vitality forms of children with ASD, suggesting that interpersonal misattunement may arise from perceptual asymmetries rather than from timing deficits alone [49]. Therefore, diminished synchrony may reflect not only individual impairments but also mismatched expressive styles between interlocutors. To disentangle synchrony impairments from mismatches in communicative style, future research should systematically manipulate speaker–listener pairings, including ASD speakers and TD listeners, and analyze the perceptual reciprocity of expressive motor dynamics.

Comment 4: It should be important to clarify that there are no differences between TD speakers in TD-TD dyads and TD-ASD dyads.

Our Response 4: Thank you for highlighting the importance of clarifying whether any differences existed between TD speakers in the TD–TD and TD–ASD conditions. We confirm that these were two separate groups of TD participants. Specifically, 36 TD individuals were assigned to the TD–TD condition, while a different group of 18 TD individuals were assigned to the TD–ASD condition. No participant took part in both conditions.

We have now clarified this point in the revised manuscript (Section 2.1, Participants) to avoid any confusion. All TD participants were undergraduate students at Kyoto University of Education. Participants in both TD conditions received identical instructions, materials, and environmental conditions. Additionally, as noted in the Procedures section, TD speakers were blind to the diagnostic status of their communication partners, minimizing any potential influence of prior assumptions on their communicative behavior (see Our Response 8 for further details).

We hope this clarification adequately addresses the reviewer’s concern and reinforces the methodological rigor of the study.

[Added Sentences, Page 3, Lines 134-136]

These two groups of TD participants were mutually exclusive, and no individual participated in both conditions. All TD participants were undergraduate students at Kyoto University of Education.

[Added Sentences, Page 6, Lines 229-230]

In both conditions, the experimental procedures and environmental settings were held constant to ensure consistency across interactions.

Comment 5: The TD and ASD groups are not reported to be IQ-matched, despite the fact that this would be important in order to rule out the possibility that a lower IQ alone could explain the anomalies in the synchrony metrics observed in the ASD group.

In the Participants section, the authors stated that clinical, cognitive, and diagnostic characteristics of participants in the ASD group were used to evaluate individual differences within the ASD group and to explore potential associations between participant profiles and synchrony performance. In the discussion section, it is raised the possibility that IQ could explain some variability, but there are no analyses. Please report analyses and clarify the discussed aspects by assessing separately the relationship between AQ and IQ with the synchronization profile.

Our Response 5: Thank you for your thoughtful comment regarding the potential influence of intellectual functioning on interpersonal synchrony in ASD. While we agree that IQ matching can be important in certain study designs, as you pointed out, it was not the focus of the current investigation.

In this study, we did not aim to examine the relationship between synchrony and individual traits such as IQ or Autism-Spectrum Quotient (AQ). Rather, our focus was on comparing interpersonal synchrony in face-to-face communication between individuals with a formal clinical diagnosis of ASD and typically developing (TD) controls. The ASD group consisted of individuals with confirmed diagnostic histories, and individuals with moderate to severe intellectual disability were excluded as part of the clinical screening process. However, specific IQ data were not collected for the TD group, and were therefore not used as a basis for group matching or analysis.

To avoid potential overinterpretation and to maintain alignment with the study’s core objectives, we have removed all content related to speculative individual variability—including the previous discussion titled Contextual and Individual Variability in ASD Synchrony Patterns. Instead, this section has been replaced with a new discussion focused on the specific methodological and theoretical issues raised in Comment 10.

Furthermore, in Section 2.1 (Participants), we have removed the previous detailed profiling—including Table 1—and any descriptive content that referenced AQ or IQ data.

We hope these revisions clarify our research focus and demonstrate our commitment to appropriately interpreting and presenting the findings within the intended scope of the study.

[Removed Sentences]

Table 1 summarizes the clinical, cognitive, and diagnostic characteristics of participants in the ASD group, including comorbidities, medication status (noted as [+] for current medication, [–] for no medication), IQ or DQ scores, and AQ scores. Comorbid conditions such as attention-deficit/hyperactivity disorder (ADHD), specific learning disorder (SLD), epilepsy, and color vision deficiency are indicated on an individual basis. For some participants, IQ or DQ values were not explicitly reported but confirmed to be within the normal range (WNR). Missing or unavailable information is marked with an em-dash (“—”). These data were used to evaluate individual differences within the ASD group and to explore potential associations between participant profiles and synchrony performance.

[Removed Table]

ASD Participant No. (Gender)

Comorbidities & Medication (+/–)

IQ/DQ

AQ

A1(F)

ADHD (-)

92

30

A2(F)

Average Range

28

A3(F)

Average Rage

30

A4(F)

97

21

A5(M)

Epilepsy (+)

87

38

A6(M)

121

33

A7(M)

102

35

A8(M)

75

34

A9(F)

ADHD (+)

87

38

A10(M)

ADHD (-)

115

36

A11(M)

SLD

86

35

A12(F)

139

30

A13 (M)

86

10

A14 (M)

Color Vision Deficiency

93

32

A15 (M)

93

22

A16 (F)

SLD, ADHD (-)

111

37

A17 (F)

97

38

A18 (F)

69

23

Table 1. Clinical, cognitive, and diagnostic characteristics of ASD participants.

Comment 6: The experimental procedure should be improved. Condition, trial, and session are not well reported. For example, in line 197, it is stated that “Prior to each trial, participants were randomly assigned as either speaker or listener.” However, it is not clear what a trial is, there seems to be only one trial in each experimental session. Please describe the procedure in more detail and be consistent with the names.

Our Response 6: We appreciate the reviewer’s observation regarding the ambiguity and inconsistency in our use of terms such as condition, trial, and session, as well as the need for greater clarity in describing the experimental procedures.

In response, we have substantially revised Section 2.3. Experimental Procedures to provide a clearer and more detailed account of the experimental task. Specifically, we have now divided the description into TD condition and ASD condition, explicitly outlining the assignment of roles, the communication task, the instructions given to each participant, and the timing and structure of the experimental interaction.

To avoid confusion, we have removed references to trial and session, which may have implied repeated or multi-phase testing. These terms have been replaced throughout the manuscript with more precise expressions such as "interaction," "task," or "experimental interaction", depending on context.

[Revised Sentences, Pages 4-6, Lines 178-241]

(Before)

2.3. Experimental Procedures

The procedure followed the paradigms developed in previous studies on head motion synchrony during communication [31]. Participants engaged in a unidirectional verbal communication task, seated face-to-face. Prior to each trial, participants were randomly assigned as either speaker or listener.

Speakers were provided with a Japanese Wikipedia article titled "Cashless Society" before the experiment and instructed to prepare to explain its contents in their own words. The article contained 2,599 characters, which was sufficient for a 5–10-minute explanation. Participants (particularly listeners) who were familiar with the topic were excluded from participation.

Speakers were instructed to maintain natural eye contact and speak clearly without deviating from the assigned topic. Listeners were not permitted to ask questions or initiate conversation, but were encouraged to demonstrate engagement through nonverbal behaviors such as nodding, facial expressions, and short vocal acknowledgments (e.g., “un,” “hai,” “ee,” which are equal to “mmhm,” “uh huh”, and “yeah” in English) [35–37]. Participants were instructed not to touch or adjust the sensors during the session.

Each session began with the experimenter announcing, "The experiment is starting," and clapping once to log a timestamp on the synchronization accelerometer. The experimenter then exited the room. Upon completing the explanation, the speaker rang a bell to indicate the end of the task. The experimenter re-entered the room and concluded the session with the statement, "The experiment is ending," accompanied by a single clap, which was again recorded by the synchronization sensor.

After the experiment, participants completed a brief questionnaire to assess subjective concentration and comprehension using the following items:

Q1: "Were you able to concentrate on the explanation?"

Q2: "Was the content of the explanation easy to understand?"

Each item was rated on a five-point semantic differential (SD) scale ranging from 1 (strongly disagree) to 5 (strongly agree).

(After)

2.3. Experimental Procedures

Participants engaged in a structured, face-to-face communication task designed to elicit naturalistic head movements under ecologically valid but experimentally controlled conditions. Each interaction involved one participant assuming the role of speaker and the other the role of listener. The communication took place in a quiet room with stable lighting and minimal environmental distraction. Participants were seated across from each other at a distance of 1.8 meters and instructed to maintain natural posture and gaze throughout the task. The primary goal was to simulate a realistic monologue-type conversation, allowing for the emergence of implicit nonverbal synchrony through nodding, subtle facial gestures, and other spontaneous body movements.

TD Condition

In the TD condition, each dyad was composed of two TD participants who were randomly assigned to the roles of speaker and listener. The speaker received a printed article titled "Cashless Society", adapted from the Japanese version of Wikipedia. This article, approximately 2,600 characters in length, was selected because it covered a contemporary but neutral topic, ensuring low emotional valence and limited prior familiarity. Speakers were given time to read and mentally prepare their explanation before the task began, but they were not allowed to take notes or rehearse aloud.

During the interaction, the speaker was asked to explain the content of the article in their own words, maintaining a natural speaking style and steady eye contact. They were encouraged to avoid reading verbatim or using exaggerated gestures. The listener was instructed not to interrupt, ask questions, or initiate conversation. However, to simulate genuine listening behavior, they were encouraged to exhibit backchannel responses such as nodding, brief affirmative sounds (e.g., "un," "hai," "ee"), and facial expressions conveying attention or understanding [35–37]. Both participants were instructed not to touch the head-mounted accelerometers or adjust their seating during the task.

To mark the beginning of the recording, the experimenter stood at the edge of the room and said, "The experiment is starting," followed by a single clap, which served as a timestamp on the synchronization sensor. The experimenter then exited the room to minimize potential influence on the participants' behavior. Upon completing the explanation, the speaker rang a small bell positioned on the table. The experimenter re-entered the room and concluded the task with the statement, "The experiment is ending," followed by another clap to mark the end of data collection. These claps were synchronized with the third accelerometer and video recording to delineate the active recording window.

ASD Condition

In the ASD condition, each dyad was composed of one TD participant and one participant with ASD. The TD participant always assumed the role of speaker, while the ASD participant served as the listener. This fixed-role design was intentional and aimed to isolate the receptive aspects of nonverbal synchrony—particularly how individuals with ASD responded to the timing and rhythm of the speaker's head movements and speech patterns.

The procedure was otherwise identical to the TD condition. The TD speaker received the same article ("Cashless Society") and followed the same instructions for preparing and delivering their explanation. The ASD listener was given clear but simple instructions using visual aids and verbal clarification when necessary, ensuring full comprehension of the task expectations. In both conditions, the experimental procedures and environmental settings were held constant to ensure consistency across interactions.

Importantly, to minimize potential expectancy effects or social bias, the speakers were not informed of the diagnostic status of their communication partners. All participants were told only that they would be paired with another person for a communication task. There was no mention of clinical background or group membership. As a result, speakers in both the TD and ASD conditions were effectively blind to the listener's diagnostic status, helping to ensure that their verbal and nonverbal behaviors were not influenced by assumptions or stereotypes.

This task structure allowed for the emergence of natural head motion synchrony, while maintaining control over the content and direction of communication. It also ensured consistency across dyads, enabling direct comparisons between groups based on movement patterns alone.

Typical examples of replacement of "trial" and "session".

(Before) A third sensor (TSND151, ATR-Promotions, Japan) served as a synchronization marker by recording timestamped signals at the start and end of each trial.

(After) A third sensor (TSND151, ATR-Promotions, Japan) served as a synchronization marker by recording timestamped signals at the start and end of each experiment.

(Before) A trained researcher remained present to monitor for signs of fatigue or discomfort throughout the session.

(After) A trained researcher remained present to monitor for signs of fatigue or discomfort throughout the experiment.

(Before) Using the distribution of detected phase differences over the course of each session, four statistical features were extracted to quantitatively characterize head motion synchrony:

(After) Using the distribution of detected phase differences over the course of each interaction, four statistical features were extracted to quantitatively characterize head motion synchrony:

(Before) Relative frequency quantifies the overall activity level of synchrony within a session, while mean phase difference captures the directional lead-lag structure between speaker and listener.

(After) Relative frequency quantifies the overall activity level of synchrony throughout the experimental interaction, while mean phase difference captures the directional lead-lag structure between speaker and listener.

[Removed Sentence]

Prior to each trial, participants were randomly assigned as either speaker or listener.

Comment 7: What about the concentration and comprehension questions? Is there any difference between TD and ASD listeners? Do they explain any variability in the listener synchronization?

Our Response 7: Thank you for your thoughtful question regarding the post-task concentration and comprehension items. While these questions were included as a brief self-check to monitor participant engagement, we decided to remove the corresponding sentences from the manuscript in order to maintain methodological focus and conceptual clarity.

[Removed Sentences]

After the experiment, participants completed a brief questionnaire to assess subjective concentration and comprehension using the following items:

Q1: "Were you able to concentrate on the explanation?"

Q2: "Was the content of the explanation easy to understand?"

Each item was rated on a five-point semantic differential (SD) scale ranging from 1 (strongly disagree) to 5 (strongly agree).

Comment 8: Were the speakers aware if the listener was a TD or ASD? Did you control this aspect?

Our Response 8:

Thank you for this important question. In our experimental design, speakers were not informed of the diagnostic status (TD or ASD) of their communication partner. All participants were introduced simply as “individuals participating in a communication study,” and no information regarding diagnostic group membership was disclosed at any point before or during the task.

This approach was taken to minimize potential expectancy effects and to ensure that the speakers' behavior—including speech delivery and nonverbal cues—was not influenced by assumptions about the listener’s clinical background. We have now clarified this point in the revised manuscript.

[Added Sentences, Page 6, Lines 231-241]

Importantly, to minimize potential expectancy effects or social bias, the speakers were not informed of the diagnostic status of their communication partners. All participants were told only that they would be paired with another person for a communication task. There was no mention of clinical background or group membership. As a result, speakers in both the TD and ASD conditions were effectively blind to the listener's diagnostic status, helping to ensure that their verbal and nonverbal behaviors were not influenced by assumptions or stereotypes.

This task structure allowed for the emergence of natural head motion synchrony, while maintaining control over the content and direction of communication. It also ensured consistency across dyads, enabling direct comparisons between groups based on movement patterns alone.

Comment 9: Data reported in Tables S1 and S2 highlight that there are much more variabilities in the session duration of the TD-ASD group than in the TD-TD group. Did you check the possible impact of different durations on synchronization measures?

Our Response 9: Thank you for your insightful comment. To assess the potential influence of experiment duration on synchronization metrics, we conducted Pearson correlation analyses between experiment duration and each synchronization measure (i.e., density, mean phase difference, standard deviation, and kurtosis). To evaluate whether session duration influenced the observed synchrony measures, we conducted Pearson correlation analyses using the combined data from both the TD and ASD groups. The results indicated no significant correlations between session duration and any of the synchronization measures:

  • Density: r = –0.0004, p = .998
  • Mean Phase Difference: r = –0.0529, p = .759
  • Standard Deviation (SD): r = –0.0348, p = .840
  • Kurtosis: r = 0.1294, p = .452

These results consistently suggest that session duration was not systematically associated with any of the synchronization metrics.

When the analysis was limited to the ASD group, we again found no significant correlations between session duration and any of the synchrony measures:

  • Nod Density: r = 0.1670, p = .508
  • Mean Phase Difference: r = –0.0761, p = .765
  • Standard Deviation (SD): r = –0.0023, p = .992
  • Kurtosis: r = 0.1170, p = .628

These results further support the interpretation that session duration did not influence the measured synchrony features, even within the ASD group. Accordingly, we have included the results of this analysis in the Results section.

[Added Sentences, Page 10, Lines 374-379]

Influence of Experiment Duration on Synchrony Measures

To examine the potential effect of experiment duration on synchrony, we conducted Pearson correlation analyses between measurement period and each synchrony feature (density, mean phase difference, SD, and kurtosis). Using the combined data from the TD and ASD groups, no significant correlations were observed: density (r = –0.0004, p = .998), mean phase difference (r = –0.0529, p = .759), SD (r = –0.0348, p = .840), and kurtosis (r = 0.1294, p = .452). These results indicate that experiment duration did not affect the synchrony measures.

Comment 10: Moreover, Pair 9 seems to be an outlier in the mean phase difference, the standard deviation, and the kurtosis. Did you check whether the data are stable without this outlier?

Our Response 10: Thank you for your thoughtful comment regarding Pair 9 in the ASD group. We agree that this pair exhibited extreme values in certain synchrony measures, particularly in the mean phase difference, standard deviation and kurtosis. To assess the potential influence of this dyad, we conducted additional analyses excluding Pair 9 from the dataset.

  • The significant group differences in density (t = 3.49, p = .001) and standard deviation (t = –3.14, p = .004) remained robust.
  • The difference in mean phase difference (t = –1.28, p = .210) remained non-significant, which is consistent with the pattern observed in the original analysis.
  • The result for kurtosis also remained marginal (p = .065).

These findings confirm that the main patterns of group differences were not driven by the influence of Pair 9, indicating that the results are stable and reliable even when the potential outlier is excluded.

In behavioral data, it is not uncommon to encounter extreme values that deviate markedly from the group distribution. However, the decision to exclude such data points must be grounded in theoretical and empirical justification, rather than based solely on statistical deviation. Outliers may reflect genuine individual differences, particularly in heterogeneous populations such as individuals with autism spectrum disorder (ASD), where greater variability in social and motor responses is well-documented (e.g., Georgescu et al., 2020; Fitzpatrick et al., 2016). Consistent with recommendations in the behavioral sciences (Wilcox, 2017; Leys et al., 2013), our approach emphasizes the importance of contextual and observational validation when evaluating atypical data, rather than relying solely on distributional assumptions. This ensures that valid, meaningful variations are not mistakenly discarded as noise.

In the present study, although one dyad in the ASD group exhibited relatively low or high values in certain synchrony metrics, we confirmed through direct behavioral observation—based on the recorded video—that the participants remained attentive and engaged throughout the interaction. The inclusion of this data point, therefore, was not only methodologically justified but also necessary to preserve the natural variability inherent in the target population.

We have now clarified this point in the revised manuscript.

[Revised Sentences, Page 13, Lines 487-505]

(Before)

Contextual and Individual Variability in ASD Synchrony Patterns

Although group-level differences were robust, notable within-group variability was observed among ASD participants. While some ASD pairs (e.g., participants A4, A10, A11, A12) demonstrated high synchrony frequency, others (e.g., A1, A5, A9) showed minimal synchrony. One pair (A11) showed high frequency but low synchrony strength. Video analyses revealed that high-synchrony ASD participants exhibited frequent backchannel feedback, whereas low-synchrony pairs rarely displayed such behavior. This suggests that frequent backchannel behaviors—such as nodding and vocal affirmations ("uh-huh", "yes")—are strongly associated with higher synchrony scores, supporting prior research linking feedback timing to interpersonal fluency [54–58]. It also raises the possibility that individuals with ASD may engage in such feedback through camouflage and masking behaviors [59,60].

Moreover, comparisons of AQ and IQ/DQ scores suggest that cognitive and neuro-psychological factors may influence synchrony outcomes. ASD participants with lower synchrony scores tended to show higher levels of autistic traits (as measured by AQ) and lower intellectual functioning (IQ/DQ), with some also receiving psychiatric medication or presenting comorbidities. These patterns indicate a possible association between synchrony characteristics and ASD symptom severity, cognitive profile, and medical status. Importantly, they highlight the need for a dimensional, rather than categorical, framework for understanding synchrony impairments—one that accounts for the heterogeneity within the ASD population. These factors warrant further investigation in studies with larger, more diverse cohorts.

(After)

Consideration of Outliers in Synchrony Data

Although one dyad (Pair ID 9) in the ASD group exhibited extreme values in certain synchrony measures—particularly mean phase difference, standard deviation, and kurtosis—the exclusion of this data point did not alter the overall pattern of results (see Table S2). Specifically, statistical group differences in synchrony activity and temporal variability remained significant, suggesting that the findings were robust and not driven by a single outlier.

Importantly, a behavioral review of the recorded interaction revealed that both participants in this dyad remained attentive and engaged throughout the task. This observation supported the inclusion of the data, as the atypical synchrony metrics likely reflected genuine individual differences rather than technical artifacts or protocol violations [22,25].

Future studies should further examine the implications of individual differences in synchrony metrics, including those that fall at the statistical extremes. Rather than treating such data as noise, it may be fruitful to explore how atypical synchrony patterns relate to specific cognitive, behavioral, or diagnostic subtypes within the autism spectrum.

Additionally, robust statistical frameworks—such as median-based or nonparametric approaches—may be applied to complement mean-based comparisons, ensuring that the analysis remains sensitive while minimizing the influence of extreme values. Combining behavioral validation with robust statistics will be crucial for improving the interpretability and generalizability of synchrony-based metrics in clinical research.

Reviewer 2 Report

Comments and Suggestions for Authors

Dear Editors,

Thank you for granting me the opportunity to review this manuscript, which presents a novel, robust, and high-resolution framework for quantifying synchrony and identifying its disruptions in ASD. I find this work quite interesting. However, I have one major concern. As the authors aim to apply this novel method in real-world clinical settings, financial considerations become a critical factor in patients’ medical decision-making processes. The manuscript does not discuss this aspect.

My concern is whether, compared to traditional methods, this new approach is cost-effective and accessible for children from migrant families and those with lower socioeconomic status. I would appreciate it if the authors could address the affordability and potential economic barriers associated with implementing this method in diverse clinical populations.

Author Response

Reviewer 2

Comment 1: Thank you for granting me the opportunity to review this manuscript, which presents a novel, robust, and high-resolution framework for quantifying synchrony and identifying its disruptions in ASD. I find this work quite interesting. However, I have one major concern. As the authors aim to apply this novel method in real-world clinical settings, financial considerations become a critical factor in patients’ medical decision-making processes. The manuscript does not discuss this aspect.

My concern is whether, compared to traditional methods, this new approach is cost-effective and accessible for children from migrant families and those with lower socioeconomic status. I would appreciate it if the authors could address the affordability and potential economic barriers associated with implementing this method in diverse clinical populations.

Our Response 1: We sincerely thank the reviewer for their thoughtful and important comment regarding the affordability and accessibility of our proposed method, especially in the context of clinical application among children from migrant families or lower socioeconomic backgrounds.

We agree that financial considerations are a critical factor in real-world medical decision-making, and we appreciate the opportunity to elaborate on this aspect. Compared to traditional neurophysiological methods such as fMRI or EEG, our approach relies on relatively low-cost, non-invasive, and commercially available technologies (e.g., wireless accelerometers and standard video recording equipment). Notably, compact accelerometer modules used for basic motion tracking are available at prices ranging from $5 to $20 USD, making this method particularly accessible for research and clinical settings with limited resources.

The experimental setup requires only a quiet room and minimal operator training, which enhances its feasibility in diverse clinical settings, including under-resourced environments. Furthermore, because the method does not require verbal responses or task-specific performance, it may be particularly well-suited for children with varying language abilities or cultural backgrounds.

To reflect this important point, we have now added a brief paragraph in the Discussion section acknowledging the economic and practical advantages of the current method, as well as its potential for scalable implementation in community and clinical settings.

We thank the reviewer once again for raising this valuable perspective.

[Revised Sentences, Pages 13-14, Lines 523-551]

(Before)

Limitations and Future Directions

Despite its contributions, this study has several limitations. First, while our use of unidirectional verbal communication allowed for controlled comparisons, it does not fully capture the complexities of natural social interactions. Future studies should examine synchrony in both dyadic and multiparty conversations involving reciprocal turn-taking to provide a more holistic understanding of interpersonal coordination. Furthermore, incorporating multimodal data encompassing multisensory input, non-verbal behavior, and physiological signals may enhance the analytical framework for investigating complex social dynamics [65–69].

Second, future work could benefit from the integration of Kinect-based motion capture systems, which offer markerless, three-dimensional tracking of full-body movements with high spatial and temporal resolution. Recent work by Kwon (2025) demonstrated the feasibility of using Kinect and phase difference analysis to detect subtle synchrony patterns in naturalistic settings, suggesting that such systems could enhance ecological validity while maintaining analytical rigor [70]. Incorporating Kinect technology may allow for a richer depiction of bodily coordination beyond head motion alone, including postural shifts and gesture synchrony.

Third, the use of machine learning techniques—such as support vector machines or logistic regression—could enable the classification of ASD based on synchrony features. By training models on features like relative frequency, standard deviation, and kurtosis, researchers may identify distinct synchrony profiles predictive of diagnostic status. This approach has already shown promise in pilot studies and could pave the way for scalable, data-driven tools to support early detection and individualized intervention strategies.

Finally, while synchrony metrics show promise as potential diagnostic markers, they should be integrated with other behavioral indicators to enhance sensitivity and specificity. Given the observed variability within the ASD group, personalized approaches to assessment and intervention may be necessary.

(After)

Limitations and Future Directions

Despite its contributions, this study has several limitations. First, while our use of unidirectional verbal communication allowed for controlled comparisons, it does not fully capture the complexities of natural social interactions. Future studies should ex-amine synchrony in both dyadic and multi-party conversations involving reciprocal turn-taking to provide a more holistic understanding of interpersonal coordination. Furthermore, incorporating multimodal data encompassing multisensory input, non-verbal behavior, and physiological signals may enhance the analytical framework for investigating complex social dynamics [67–71].

Second, in contrast to traditional neuroimaging or neurophysiological approaches, the present method is based on non-invasive, portable, and relatively low-cost technologies such as accelerometers and video cameras. This simplicity enhances its feasibility for implementation in clinical environments with limited resources. However, systematic cost–benefit evaluations and pilot implementations in real-world clinical settings are warranted to assess feasibility across diverse populations. To expand its utility, future studies should explore how this framework can be practically adapted and scaled for use in diverse real-world clinical settings, particularly to improve accessibility for children with limited healthcare access. Specifically, we should aim to adapt and field-test this framework in community-based clinical contexts, such as public health centers, school-based screening programs, or mobile outreach services.

Additionally, simplified sensor setups (e.g., single-sensor configurations) and automated analysis pipelines could be developed to further reduce cost and technical barriers to adoption. In this regard, ongoing work is already investigating the use of camera-based systems for real-time detection and visualization of head motion synchrony, with the goal of eliminating the need for wearable sensors and facilitating seamless integration into clinical settings. This direction is outlined in our previous study, which introduces a practical visualization system for real-time synchrony detection using RGB video input [72]. Such innovations are expected to enhance the accessibility and scalability of synchrony-based screening tools in everyday healthcare environments.

Round 2

Reviewer 1 Report

Comments and Suggestions for Authors

I did not appreciate that the authors decided to remove from the paper Table 1 (it should at least be included in the supplementary). I would also expect a more in-depth discussion of the impact of IQ, especially since the previous version of the manuscript suggested it may play a role. Given that ASD and TD participants are not IQ-matched, understanding the potential influence of IQ is essential, as it could affect the interpretation of the results.

Additionally, I am concerned that, rather than addressing my previous questions: “What about the concentration and comprehension questions? Is there any difference between TD and ASD listeners? Do they explain any variability in listener synchronization?”, the authors chose to remove the sentence informing the reader that these questions were asked. These aspects could be relevant to interpreting group differences. To conclude that there is a robust impairment in nonverbal head motion synchrony in ASD, possible confounds should be controlled.

Moreover, I believe it is important to demonstrate the absence of differences in how TD speakers interact with ASD vs TD listeners to support the stated conclusions. For example, it can be tested whether there are differences in the density of nodding of TD speakers in the TD condition vs. the density of nodding of TD speakers in the ASD condition. If differences in TD speakers between conditions emerge, it would suggest that the synchrony observed between TD-ASD pairs differs from that between two TD pairs, not solely that there is an impairment of ASD participants. To conclude that there is an ASD-specific impairment, it is essential to show that TD speakers behave consistently regardless of whether they are interacting with a TD or ASD listener.

Other small aspects:

There is no reference to Table S1 in the main text.

I would insert the analysis without couple 9 in Supplementary

It is unclear whether there was someone in the room to control “A trained researcher remained present to monitor for signs of fatigue or discomfort throughout the experiment” or not “The experimenter then exited the room to minimize potential influence on the participants' behavior”

Line 314, the longer tail seems to be the negative one

Line 324, I do not think it is correct to refer to TD-ASD pairs as “ASD pairs”

Author Response

Reviewer 1

Comments and Suggestions for Authors

Comment 1: I did not appreciate that the authors decided to remove from the paper Table 1 (it should at least be included in the supplementary). I would also expect a more in-depth discussion of the impact of IQ, especially since the previous version of the manuscript suggested it may play a role. Given that ASD and TD participants are not IQ-matched, understanding the potential influence of IQ is essential, as it could affect the interpretation of the results.

Our Response 1: We appreciate the reviewer’s insightful comment regarding the role of IQ in our study. As noted, IQ data were collected only for the ASD group as part of the clinical diagnostic process, and thus, the TD group was not explicitly matched on this variable. We acknowledge that this represents a limitation in our study design. To address this, we have expanded the Discussion section to consider the potential influence of intellectual functioning on our findings. Specifically, we reference prior research indicating that cognitive abilities can affect social interaction patterns in individuals with ASD. We have also emphasized the importance of including comprehensive cognitive assessments in future studies to better understand the interplay between IQ and social synchrony in ASD.

Additionally, our previous revision may have misunderstood the intent of this comment. While our initial focus was on removing diagnostic and demographic tables to simplify the manuscript, we acknowledge that the potential influence of individual differences—including cognitive profiles such as IQ or AQ—warrants deeper attention. Therefore, we restored Table 1 and the accompanying descriptive information to clarify individual-level characteristics within the ASD group.

To address this point more thoroughly, we conducted additional analyses using Autism-Spectrum Quotient (AQ) scores that were collected for all participants. We compared the AQ scores of TD speakers across the two conditions (TD–TD vs. TD–ASD) and also compared TD listeners and ASD listeners. Results showed no significant difference in AQ scores between TD speakers in the two conditions (t(34) = 0.25, p = 0.804), suggesting that TD participants’ cognitive traits were comparable across conditions. However, AQ scores were significantly higher for ASD listeners compared to TD listeners (t(34) = –6.55, p < 0.0001), as expected given diagnostic status. These analyses and the associated data have been added to the revised manuscript (see Results Section) and are summarized in Supplementary Table S3.

[Added Sentences, Page 12, Lines 412-426]

Autism-Spectrum Quotient (AQ) Comparisons

To examine whether individual differences in autistic traits could account for group differences in head motion synchrony, we analyzed Autism-Spectrum Quotient (AQ) scores for all participants. First, we compared the AQ scores of TD speakers assigned to the TD condition and those assigned to the ASD condition. The mean AQ scores were 16.56 (SD = 6.17) for the TD condition and 15.94 (SD = 8.36) for the ASD condition. An independent samples t-test indicated no significant difference between these groups (t(34) = 0.25, p = 0.804), suggesting that the TD speakers across conditions were similar in their autistic trait profiles.

In contrast, ASD listeners showed substantially higher AQ scores (M = 30.56, SD = 7.47) than TD listeners (M = 15.56, SD = 6.21). This difference was statistically significant (t(34) = –6.55, p < 0.0001), reflecting the diagnostic distinction between groups. These findings confirm that AQ traits among TD participants were balanced across conditions, and that group-level differences in synchrony are unlikely to be driven by AQ variation in the TD population. Full AQ data are provided in Supplementary Table S3.

[Added Sentences, Page 15, Lines 557-593]

Trait-Level Explanations for Reduced Synchrony in ASD: The Roles of AQ and IQ

The current analysis of Autism-Spectrum Quotient (AQ) scores helps clarify the role of individual differences in autistic traits in shaping group-level differences in head motion synchrony. Our findings show that TD speakers assigned to the TD and ASD conditions did not differ significantly in their AQ scores, indicating that autistic trait levels were balanced across the two experimental groups. In addition, to examine whether TD speakers modulated their behavior based on the diagnostic status of their communication partner, we compared head movement density between the TD and ASD conditions. This analysis revealed no significant differences between conditions, suggesting that TD speakers exhibited consistent levels of nonverbal behavior regardless of their partner's diagnostic background. These findings support the interpretation that the reduced synchrony observed in TD–ASD dyads cannot be attributed to variation in autistic traits or behavioral modulation among TD participants.

As expected, AQ scores were significantly higher among ASD listeners than TD listeners, consistent with their diagnostic status. This robust group difference confirms the integrity of the diagnostic contrast and suggests that elevated autistic traits in the ASD group are a core component of the observed reduction in nonverbal synchrony. Taken together, these findings strengthen the argument that reduced head motion synchrony in ASD is not an artifact of sample-level trait heterogeneity, but rather reflects a meaningful social-communicative difference characteristic of autism. Future studies may benefit from examining how specific AQ subdomains—such as attention to detail or social skills—relate to different dimensions of nonverbal synchrony.

One limitation of the present study is that IQ data were available only for the ASD group, and thus the two groups were not explicitly matched for intellectual functioning. Previous studies have shown that even in samples where ASD and TD groups were matched on IQ, or where all participants had average or above-average intelligence, individuals with ASD consistently exhibited impairments in nonverbal synchrony and social coordination [22,25]. These findings support the interpretation that synchrony deficits in ASD reflect a core feature of the condition rather than being secondary to general cognitive delay. However, other research has shown that motor performance is positively correlated with IQ in individuals with ASD [56]. In particular, children with lower IQ exhibited poorer gross motor skills and made more mirroring errors compared to those with higher IQ levels [56]. These findings suggest that IQ may modulate the severity of motor coordination difficulties in ASD. Therefore, future research would benefit from including comprehensive cognitive assessments in both groups to better isolate the respective contributions of intellectual functioning and autism-specific impairments in nonverbal coordination.

  1. Fitzpatrick, P.; Frazier, J.A.; Cochran, D.M.; Mitchell, T.; Coleman, C.; Schmidt, E.R. Impairments of social motor synchrony evident in autism spectrum disorder. Front. Psychol. 2016, 7(1323), 1-13.
  2. Georgescu, A.L.; Koeroglu, S.; Hamilton, A.F.D.C.; Vogeley, K.; Falter-Wagner, C.M.; Tschacher, W. Reduced nonverbal interpersonal synchrony in autism spectrum disorder independent of partner diagnosis: A motion energy study. Mol. Autism 2020, 11, 1–14. https://doi.org/10.1186/s13229-019-0305-1
  3. Kaur, M.; Srinivasan, S.M.; Bhat, A.N. Comparing motor performance, praxis, coordination, and interpersonal synchrony between children with and without Autism Spectrum Disorder (ASD). Res. Dev. Disabil. 2018, 72, 79–95.

[Added Table]

Table S3. Autism-Spectrum Quotient (AQ) Scores of Speakers and Listeners in the TD and ASD Conditions

Pair ID

TD Condition

ASD Condition

TD speaker

TD Listener

TD speaker

ASD Listener

1

10

12

21

30

2

12

15

7

28

3

27

7

22

30

4

13

21

10

21

5

21

17

8

38

6

19

21

28

33

7

17

11

29

35

8

20

24

15

34

9

9

15

13

38

10

19

20

13

36

11

28

16

14

35

12

15

7

26

30

13

6

23

29

10

14

22

7

13

32

15

20

7

19

22

16

16

12

12

37

17

16

20

1

38

18

8

25

7

23

Comment 2: Additionally, I am concerned that, rather than addressing my previous questions: “What about the concentration and comprehension questions? Is there any difference between TD and ASD listeners? Do they explain any variability in listener synchronization?”, the authors chose to remove the sentence informing the reader that these questions were asked. These aspects could be relevant to interpreting group differences. To conclude that there is a robust impairment in nonverbal head motion synchrony in ASD, possible confounds should be controlled.

Our Response 2: Thank you for revisiting this important point. In response to your concern, we have now conducted statistical comparisons of both concentration and comprehension ratings between TD and ASD listeners. These ratings were originally collected through post-task self-report items using a five-point Likert scale.

For concentration, the mean score was 3.28 (SD = 1.32) for TD listeners and 3.44 (SD = 1.29) for ASD listeners. For comprehension, the mean score was 3.44 (SD = 1.15) for TD listeners and 3.94 (SD = 1.16) for ASD listeners. Mann–Whitney U tests revealed no statistically significant group differences in either concentration (U = 173.5, p = 0.720) or comprehension (U = 206.0, p = 0.149). These results suggest that differences in self-reported engagement or task understanding are unlikely to explain the observed group-level differences in nonverbal synchrony. The corresponding results have been reported in the revised manuscript (see Result Section).

[Added Sentences, Page 12, Lines 439-449]

Listener Engagement and Comprehension

To address whether differences in listener engagement may account for the observed group-level differences in synchrony, we compared post-task self-reported ratings of concentration and comprehension between TD and ASD listeners. Each item was rated on a five-point Likert scale ranging from 1 (strongly disagree) to 5 (strongly agree). Concentration ratings were similar between groups, with TD listeners reporting a mean score of 3.28 (SD = 1.32) and ASD listeners 3.44 (SD = 1.29). Comprehension scores were also comparable, with means of 3.44 (SD = 1.15) for TD listeners and 3.94 (SD = 1.16) for ASD listeners. Mann–Whitney U tests indicated no statistically significant differences for either concentration (U = 173.5, p = 0.720) or comprehension (U = 206.0, p = 0.149). These findings suggest that listener-level differences in self-perceived engagement or understanding do not account for the reduced synchrony observed in ASD dyads.

Comment 3: Moreover, I believe it is important to demonstrate the absence of differences in how TD speakers interact with ASD vs TD listeners to support the stated conclusions. For example, it can be tested whether there are differences in the density of nodding of TD speakers in the TD condition vs. the density of nodding of TD speakers in the ASD condition. If differences in TD speakers between conditions emerge, it would suggest that the synchrony observed between TD-ASD pairs differs from that between two TD pairs, not solely that there is an impairment of ASD participants. To conclude that there is an ASD-specific impairment, it is essential to show that TD speakers behave consistently regardless of whether they are interacting with a TD or ASD listener.

Our Response 3: We thank the reviewer for this important and constructive suggestion. In response, we conducted an additional analysis comparing the head movement density of TD speakers across the two experimental conditions: when interacting with a TD listener (TD–TD dyads) and with an ASD listener (TD–ASD dyads). The mean density was 23.30 nods per minute (SD = 4.83) for the TD condition and 24.27 (SD = 5.17) for the ASD condition. An independent samples t-test revealed no statistically significant difference between conditions (t(34) = –0.56, p = 0.576, Cohen’s d = –0.19).

This result suggests that TD speakers exhibited consistent levels of nonverbal activity regardless of their listener’s diagnostic status. Accordingly, it supports the interpretation that the reduced synchrony observed in TD–ASD dyads is unlikely to be attributable to changes in TD speaker behavior, but instead reflects ASD-specific impairments in nonverbal coordination. We have added this result to the revised manuscript in Section 4.3 (“TD Speaker Behavior Across Conditions”), and detailed data are provided in the Supplementary Material (Table S4).

[Added Sentences, Page 12, Lines 427-438]

TD Speaker Behavior Across Conditions

To examine whether TD speakers behaved differently depending on the diagnostic status of their communication partner, we compared the density of head movements in the TD and ASD conditions. The mean head movement density was 23.30 nods per minute (SD = 4.83) for TD speakers in the TD condition and 24.27 nods per minute (SD = 5.17) in the ASD condition. An independent samples t-test revealed no significant difference between conditions, t(34) = –0.56, p = 0.576, Cohen’s d = –0.19.

These results suggest that TD speakers exhibited comparable nonverbal behavior across both conditions, supporting the interpretation that the reduced synchrony observed in TD–ASD dyads is primarily attributable to characteristics of the ASD participants rather than differences in TD speaker behavior. Pairwise statistics are provided in Supplementary Table S4.

Table S4. Head Movement Density of TD Speakers Across the Two Experimental Conditions (Nods/Min)

Pair ID

TD Condition

ASD Condition

1

20.3

22.8

2

25.5

20.3

3

20.7

14.9

4

21.8

26.5

5

32.5

15.1

6

26.6

29.2

7

26.1

30.6

8

25.2

28.2

9

23.4

31.3

10

21.5

25.1

11

21.4

20.1

12

21.8

30.2

13

26.6

20.1

14

28.5

31.2

15

22.2

18.3

16

17.4

24.9

17

28.2

24.9

18

9.7

23.2

Other small aspects:

Comment 4 to 8: We appreciate the reviewer’s attention to detail and have carefully addressed each of the small but important points raised in Comments 4 through 8.

Comment 4: There is no reference to Table S1 in the main text.

Our Response 4:

[Revised Sentences, Pages 9, Lines 358-359]

(Before) Detailed pair-level data for each group are provided in the Supplementary Materials.

(After) Detailed pair-level data for each group are provided in the Supplementary Materials (see Tables S1 and S2).

Comment 5: I would insert the analysis without couple 9 in Supplementary

Our Response 5:

[Added Table]

Table S5. Robustness Check of Synchrony Metrics After Excluding Pair 9 from the ASD Condition

Metric

Group Difference (t)

p-value

Significance

Density

3.49

0.001

Significant

Mean Phase Difference

–1.28

0.210

Not significant

Standard Deviation

–3.14

0.004

Significant

Kurtosis

1.91

0.065

Marginal

Note: These results reflect the group comparisons after excluding Pair 9 from the ASD condition. Findings for density and variability (SD) remained statistically significant, while the group difference in mean phase difference continued to be non-significant, consistent with the original analysis. The marginal result for kurtosis should be interpreted with caution.

[Revised Sentences, Pages 15, Lines 598-600]

(Before) Specifically, statistical group differences in synchrony activity and temporal variability remained significant, suggesting that the findings were robust and not driven by a single outlier.

(After) Specifically, statistical group differences in synchrony activity and temporal variability remained significant, suggesting that the findings were robust and not driven by a single outlier (see Table S5).

Comment 6: It is unclear whether there was someone in the room to control “A trained researcher remained present to monitor for signs of fatigue or discomfort throughout the experiment” or not “The experimenter then exited the room to minimize potential influence on the participants' behavior”

Our Response 6:

[Removed Sentence]

A trained researcher remained present to monitor for signs of fatigue or discomfort throughout the experiment.

[Added Sentences, Pages 6, Lines 239-242]

A trained researcher monitored participants for signs of fatigue or discomfort and provided clarification if needed throughout the entire experiment. The task was continuously monitored via live audio and video feed from an adjacent control room to ensure participant safety without disrupting the natural flow of interaction.

Comment 7: Line 314, the longer tail seems to be the negative one

Our Response 7:

[Revised Sentences, Pages 9, Lines 337-340]

(Before) The distribution showed long tails extending beyond ±500 ms, particularly toward positive lags, indicating delayed and inconsistent listener responses.

(After) The distribution showed long tails extending beyond ±500 ms, with a particularly extended tail toward negative lags. Since negative lags reflect listener movements that precede the speaker’s behavior, this may indicate inconsistent listener responses.

Comment 8: Line 324, I do not think it is correct to refer to TD-ASD pairs as “ASD pairs”

Our Response 8:

[Revised Sentences, Pages 9, Lines 348-350]

(Before) Together, these findings support the interpretation that TD pairs exhibit stronger and more temporally precise synchrony, while ASD pairs show weaker and temporally diffuse coordination.

(After) Together, these findings support the interpretation that TD–TD pairs exhibit stronger and more temporally precise synchrony, while TD–ASD pairs show weaker and temporally diffuse coordination.

[Revised Sentences, Pages 13, Lines 503-305]

(Before) Consistent with prior research emphasizing the role of sensorimotor coupling in social bonding [11–14,40,41], we found that TD pairs exhibited significantly higher relative frequencies of synchronized head movements compared to ASD pairs.

(After) Consistent with prior research emphasizing the role of sensorimotor coupling in social bonding [11–14,40,41], we found that TD–TD pairs exhibited significantly higher relative frequencies of synchronized head movements compared to TD–ASD pairs.

Reviewer 2 Report

Comments and Suggestions for Authors

I have no more question at this stage

Author Response

We sincerely thank the reviewer for their thoughtful evaluation and positive appraisal of our revised manuscript. We are grateful for the constructive feedback provided during the earlier stages of the review process, which helped improve the clarity and rigor of our work. We appreciate your time and consideration.

Round 3

Reviewer 1 Report

Comments and Suggestions for Authors

I appreciate the authors' efforts and the manner in which the comments were addressed. In my view, the paper has significantly improved.